

# Multi-approach gravity field models from Swarm GPS data

João Teixeira da Encarnação[1,2], Pieter Visser[1], Daniel Arnold[3], Aleš Bezděk[4], Eelco Doornbos[5], Matthias Ellmer[6], Junyi Guo[8], Jose van den IJssel[1], Elisabetta Iorfida[1], Adrian Jäggi[3], Jaroslav Klokočník[4], Sandro Krauss[7], Xinyuan Mao[3], Torsten Mayer-Gürr[7], Ulrich Meyer[3], Josef Sebera[4], C.K. Shum[8,9], Chaoyang Zhang[8], Yu Zhang[8], and Christoph Dahle[10]

[1]Delft University of Technology, Faculty of Aerospace Engineering, Kluyverweg 1, 2629 HS, Delft, The Netherlands
[2]Center for Space Research, The University of Texas at Austin, 3925 West Braker Lane, Suite 200 Austin, Texas, USA
[3]Astronomical Institute of the University of Bern, Sidlerstrasse 5, 3012 Bern, Switzerland
[4]Astronomical Institute of the Czech Academy of Sciences, Fričova 298, 251 65 Ondřejov, Czech Republic
[5]Royal Netherlands Meteorological Institute, Utrechtseweg 297, 3731 GA De Bilt, the Netherlands
[6]Jet Propulsion Laboratory, 4800 Oak Grove Drive, Pasadena, CA 91109, U.S.A.
[7]Institute of Geodesy of the Graz University of Technology, Steyergasse 30/III, 8010 Graz, Austria
[8]School of Earth Sciences of The Ohio State University, 125 Oval Dr S, Columbus, OH 43210, USA
[9]Institute of Geodesy and Geophysics, Chinese Academy of Sciences, Wuhan, China
[10]GFZ German Research Centre for Geosciences, Potsdam, Germany

**Correspondence:** João Teixeira da Encarnação (J.G.deTeixeiradaEncarnacao@tudelft.nl)

**Abstract.**

Although the knowledge of the gravity of the Earth has improved considerably with CHAMP, GRACE and GOCE satellite missions, the geophysical community has identified the need for the continued monitoring of the time-variable component with the purpose of estimating the hydrological and glaciological yearly cycles and long-term trends. Currently, the GRACE-FO

satellites are the sole dedicated provider of these data, while previously the GRACE mission fulfilled that role for 15 years. There is a data gap spanning from July 2017 to May 2018 between the end of the GRACE mission and start the of GRACE-FO, while the Swarm satellites have collected gravimetric data with their GPS receivers since December 2013.

We present high-quality gravity field models from Swarm data that constitute an alternative and independent source of gravimetric data, which could help alleviate the consequences of the 10-month gap between GRACE and GRACE-FO, as well

as the short gaps in the existing GRACE and GRACE-FO monthly time series.

The geodetic community has realized that the combination of different gravity field solutions is superior to any individual model and set up a Combination Service of Time-variable Gravity Fields (COST-G) under the umbrella of the International Gravity Field Service (IGFS), part of the International Association of Geodesy (IAG). We exploit this fact and deliver to the highest quality monthly-independent gravity field models, resulting from the combination of four different gravity field

estimation approaches. All solutions are unconstrained and estimated independently from month to month.

We tested the added value of including Kinematic Baselines (KBs) in our estimation of Gravity Field Models (GFMs) and conclude that there is no significant improvement. The non-gravitational accelerations measured by the accelerometer on-board Swarm-C were also included in our processing to determine if this would improve the quality of the GFMs, but observed that





is only the case when the amplitude of the non-gravitational accelerations is higher than during the current quiet period in solar

activity.

Using GRACE data for comparison, we demonstrate that the geophysical signal in the Swarm gravity field models is largely restricted to Spherical Harmonic degrees below 12. A $750\mathrm{km}$ smoothing radius is suitable to retrieve the temporal variations of Earth's gravity field over land areas since mid-2015 with roughly $4\mathrm{cm}$ Equivalent Water Height (EWH) agreement with respect to a GRACE-derived parametric model. Over ocean areas, we illustrate that a more intense smoothing with $3000\mathrm{km}$

radius is necessary to resolve large scale gravity variations, which agree with the aforementioned parametric model under $2\mathrm{cm}$ EWH, while at these spatial scales the model represents variations with amplitudes between 2 and $3.5\mathrm{cm}$ EWH. The agreement with GRACE and GRACE-FO over nine selected large basins under analyses is $1.19\mathrm{cm}$, $0.60\mathrm{cm/year}$ and 0.75 in terms of temporal mean, trend and correlation coefficient, respectively.

The Swarm monthly models are distributed on a quarterly basis at ESA's Earth Online Swarm Data Access (at https:

//swarm-diss.eo.esa.int/#swarm, follow *Level2longterm* and then *EGF*) and at the International Centre for Global Earth Models (http://icgem.gfz-potsdam.de/series/03_COST-G/Swarm), as well as identified with the DOI 10.5880/ICGEM.2019.006 (Encarnacao et al., 2019).

## 1 Introduction

Swarm is the fifth Earth Explorer mission by European Space Agency (ESA), launched on 22 November 2013 (Haagmans,

2004; Friis-Christensen et al., 2008). Its primary objective is to provide the best ever survey of the Earth's magnetic field and its temporal variations as well as the electric field of the atmosphere (Olsen et al., 2013). Swarm consists of three identical satellites, two flying in a pendulum formation (side-by-side, converging near the poles) at an initial altitude of about $470\mathrm{km}$ and one at an altitude of about $520\mathrm{km}$, all in near-polar orbit. In addition to a sophisticated instrument suite for observing the geomagnetic and electric field, the Swarm satellites are equipped with high-precision, dual-frequency Global Positioning

System (GPS) receivers, star trackers and accelerometers. Many recent studies and activities have shown the feasibility of observing the Earth's gravity field and its long-wavelength temporal variations with high-quality GPS receivers on board of Low-Earth Orbit (LEO) satellites (Zehentner and Mayer-Gürr, 2014; Bezděk et al., 2016; Dahle et al., 2017). For Swarm, Teixeira da Encarnação et al. (2016) successfully demonstrated the observation of long-wavelength temporal gravity. They produced solutions by three different approaches and showed that their combination resulted in improved observability of time

variable gravity, a principle that has been suggested in the frame of the initiative of the European Gravity Service for Improved Emergency Management (EGSIEM) (Jäggi et al., 2019) and demonstrated for Gravity Recovery And Climate Experiment (GRACE)-based gravity field solutions (Jean et al., 2018).

An important driver for producing LEO GPS-based gravity field solutions is to guarantee long-term observation of mass transport in the Earth system. The geophysical community has identified the need for continued monitoring of time variable

gravity for estimating the hydrological and glaciological yearly cycles and long-term trends (Abdalati et al., 2018). The US/ German GRACE mission (Tapley et al., 2004) was by far the most important space-borne global provider of the needed data





for the period from April 2002 until July 2017. GRACE Follow On (GRACE-FO) was launched in May 2018 and is expected to continue the high-quality observation of Earth's time variable gravity field for at least 5 years (Flechtner et al., 2016). Thus a time gap exists between the GRACE and GRACE-FO missions and, importantly, no missions have yet been selected for the post GRACE-FO period. It can thus be claimed that the only guarantee for sustained observation of time variable gravity from space is constituted by space-borne GPS receivers on LEO satellites. Moreover, the associated data can be used to fill the gap between the GRACE and GRACE-FO missions (be it with a different quality in terms of spatial and temporal resolution).

The studies described in this paper aimed at improving Swarm-based observation of long-wavelength time variable gravity in preparation for the operational delivery of monthly Swarm-based gravity field solutions. It is a continuation of the activities described in (Teixeira da Encarnação et al., 2016), which included the production of gravity field solutions using three different methods, referred to as Celestial Mechanics Approach (CMA) (Beutler et al., 2010), Decorrelated Acceleration Approach (DAA) (Bezděk et al., 2014), and Short-Arcs Approach (SAA) (Mayer-Gürr, 2006). In this work, a fourth method, referred to as Improved Energy Balance Approach (IEBA) (Shang et al., 2015), is added. The combination of the four gravity field solutions into combined models will be more advanced than in (Teixeira da Encarnação et al., 2016), where a straightforward averaging was applied. In the results presented in this work, the weights are derived from Variance Component Estimation (VCE) in analogy to Jean et al. (2018), in order to arrive as close as possible to statistically-optimal combined solutions, given the available combinations strategies, as described in Section 2.5.

The nominal gravity field solutions will be based on Kinematic Orbit (KO) solutions, which consist of time series of position coordinates. These time series can be considered as a condensed form of the original GPS High-low Satellite-to-Satellite tracking (hl-SST) observations, with no effect from dynamic models for the LEO satellites (the positions of the GPS satellite themselves are based on dynamic models, as usual). Three different KO solutions are produced by Delft University of Technology (TUD), Astronomical Institute of the University of Bern (AIUB), and the Institute of Geodesy Graz (IfG) of the Graz University of Technology (TUG) (van den IJssel et al., 2015; Jäggi et al., 2016; Zehentner, 2016).

We also tested another potential innovation that could conceptually lead to improved gravity field solutions, that is the use of kinematically derived baselines for the two Swarm satellites flying in a pendulum formation. Kinematic Baselines (KBs) between two LEO formation flying spacecraft can typically be derived with much better precision than the absolute positions by making use of ambiguity fixing schemes and due to cancellation of common errors (Kroes, 2006; Allende-Alba et al., 2017). The possible added value of KBs for the observation of temporal gravity field variations will be assessed making use of two different KB solutions by TUD (Mao et al., 2017) and the AIUB (Jäggi et al., 2007, 2009).

We also present a comparison of the quality of gravity field retrievals from Swarm-C observational data making use of either the available accelerometer product for this satellite (Doornbos et al., 2015) or two different non-gravitational acceleration force models.

This paper is organized as follows. More details about the methodology are provided in Section 2. Results are included and discussed in Section 3. A summary, conclusions, and outlook are given in Section 4.



For the sake of brevity, we will refer to GRACE and GRACE-FO data simply as *GRACE* data, unless there is the need to be more specific. We also interchangeably use the terms *solution* (when relevant to a set of Stokes coefficients) and *Gravity Field Model (GFM)*.

The operational activities currently under way pertaining to the combined models described in this article are conducted in the frame of the Combination Service of Time-variable Gravity Fields (COST-G), under the umbrella of International Association of Geodesy (IAG)'s International Gravity Field Service (IGFS) (Jäggi et al., 2019), with additional support from the Swarm Data, Innovation and Science Cluster (DISC) and funded by ESA. The Swarm monthly models are distributed on a quarterly basis at ESA's Earth Online Swarm Data Access (at https://swarm-diss.eo.esa.int/#swarm, follow *Level2longterm* and then *EGF*) and at the International Centre for Global Earth Models (http://icgem.gfz-potsdam.de/series/03_COST-G/Swarm), as well as identified with the DOI 10.5880/ICGEM.2019.006 (Encarnacao et al., 2019).

## 2 Methodology

In this work, we mainly intend to present the capabilities of the Swarm GFMs, in terms of their particularities and data quality, and typically refer to the relevant methodology in supporting literature. Nevertheless, this section discusses briefly some aspects of the various stages in the processing of the models, their combination and, to better prepare the discussion of results in Section 3, the approach used in the analysis of the Swarm GFMs.

### 2.1 Kinematic Orbits

The KOs are the observables from which the GFMs are estimated, since they are solely derived from the geometric distance relative to the GPS satellites. The different KO solutions are conceptually estimated in similar ways, but with the processing strategies described in detail in the references of Table 1. Furthermore, each Analysis Center (AC) makes their own choices regarding the numerous assumptions and processing options for deriving their individual KO solutions, as listed in Appendix A. The reason for the different KO solutions is to provide various options for the ACs's individual GFMs processing (see Section 2.4) and, in this way, reduce the impact of possible KO-driven systematic errors in the combined GFMs. It also enables the ACs to select which KO solution is more advantageous to the quality of their GFMs; consider that our gravity estimation approaches may be differently sensitive to the error spectra of the various KO solutions, or have different requirements on the quality of the variance-covariance information provided with the kinematic positions. This selection is done at each AC and outside the scope of the current study.

### 2.2 Kinematic Baselines

We investigate the added value of KBs in the quality of the Swarm GFMs, as presented in Section 3.1. The KB solutions, much in the same way as the KOs, are conceptually computed similarly, where fixing ambiguities is a necessary processing step to achieve the highest possible precision of the derived baselines. This constitutes the main motivation to include KBs in





**Table 1.** Overview of the Kinematic Orbits and the software packages used to estimate them

| Institute | Software | Reference |
|-----------|----------|-----------|
| AIUB | Bernese v5.3 (Dach et al., 2015; Jäggi et al., 2006) | Jäggi et al. (2016)[1] |
| IfG | Gravity Recovery Object Oriented Programming System (GROOPS) (in-house development) | Zehentner and Mayer-Gürr (2016)[2] |
| TUD | GPS High precision Orbit determination Software Tool (GHOST) (van Helleputte, 2004; Wermuth et al., 2010) | van den IJssel et al. (2015)[3] |

[1] ftp://ftp.aiub.unibe.ch/leo_orbits/swarm  [2] ftp://ftp.tugraz.at/outgoing/ITSG/tvgogo/orbits/Swarm
[3] http://earth.esa.int/web/guest/swarm/data-access

the estimation of the Swarm GFMs. The interested reader can find details in the references of Table 2; the main processing assumptions are listed in Appendix B and brief descriptions follow.

**Table 2.** Overview of the Kinematic Baselines and the software packages used to estimate them

| Institute | Software | Reference |
|-----------|----------|-----------|
| AIUB | Bernese v5.3 (Dach et al., 2015) | Jäggi et al. (2007, 2009) |
| TUD | Multiple satellites Orbit Determination using Kalman filtering (MODK) (van Barneveld, 2012) | Mao et al. (2018) |

### 2.2.1 KBs produced at AIUB

Kinematic and reduced-dynamic baselines are determined according to the procedures described by Jäggi et al. (2007, 2009, 2012). The positions of one satellite (Swarm-A) are kept fixed to a reduced-dynamic solution generated from Zero-differenced

(ZD) ionosphere-free GPS carrier phase observations. Reduced-dynamic orbit parameters of the other satellite (Swarm-C) are estimated by processing Double-differenced (DD) ionosphere-free GPS carrier phase observations with DD ambiguities resolved to their integer values. First, the Melbourne-Wübbena linear combination is analysed to resolve the wide-lane ambiguities, which are subsequently introduced as known to resolve the narrow-lane ambiguities together with the reduced-dynamic baseline determination. For the KB estimation, the same procedure may be used but it turned out to be more robust to in-

troduce the resolved ambiguities from the available reduced-dynamic baselines and not to make an attempt to independently





fix carrier phase ambiguities in the KB processing. Exactly the same carrier phase ambiguities are therefore fixed in both the reduced-dynamic and the kinematic baseline determination.

### 2.2.2 KBs produced at TUD

We take advantage of a forward and backward Extended Kalman Filter (EKF) that is run iteratively. The EKF initially runs
from the first epoch to the last epoch of each 24-hours orbit arc with 5 second step. The estimated float ambiguities and the corresponding covariance matrices (which are recorded for each epoch) are used by the Least-squares Ambiguity Decorrelation Adjustment (LAMBDA) algorithm in order to fix the maximum number of integer ambiguities (subset approach). The EKF smooths both solutions according to the bi-directional covariance matrices recorded at each epoch. In the next iteration, the smoothed orbit and fixed ambiguities are set as input and it is attempted to fix more ambiguities. The procedure
is repeated until no new integer ambiguities are fixed.

After the convergence of the reduced-dynamic baseline, a KB solution is produced using the Least-Squares (LS) method. To this purpose, the same GPS observations and fixed integer ambiguities on the two frequencies are used, where one satellite (Swarm-A) is kept fixed at the reduced-dynamic baseline solution. At least 5 observations are required on each frequency to form a good geometry. To minimize the influence of wrongly fixed ambiguities and residual outliers, a threshold of 2-sigma
of the carrier phase residual STD is set, which results in eliminating around $5\%$ data, on average. A further screening of $3\mathrm{cm}$ is set to the RMS of the kinematic baseline carrier phase observation residual. This makes it possible to screen out the epochs that are influenced by wrongly fixed ambiguities and bad phase observations. The kinematic baseline determination is also run bi-directionally to compute two solutions that are averaged according to the epoch-wise covariance matrices.

### 2.2.3 Inclusion of KBs in the estimation of Swarm GFMs

We exploit the Variational Equations Approach (VEA) (Montenbruck and Gill, 2000) implemented at IfG in the inversion of gravity field considering both KOs and KBs. The VEA and its application to KOs and KBs corresponds to the processing scheme used for the production of the ITSG-GRACE2016 (Klinger et al., 2016).

We selected a number of suitable test months with varying data quality, meeting the following criteria: GRACE monthly solutions are available for validation purposes; months with *good* GPS data quality are included as well as months with *bad* data
quality; and some months should overlap with the test months selected in the non-gravitational acceleration study (Section 2.3) for the accelerometer data tests.

The descriptions *good* and *bad* data quality refer to several issues in the context of Swarm GPS data. *Good* means that an error found in the Receiver Independent Exchange (RINEX) converter is solved (fixed since 12. April 2016), the settings of the receiver tracking loop bandwidths are optimized (several changes during lifetime), and the ionospheric activity is at a low
level. In contrast, the *bad* data hold for time periods for which these issues are not solved and the ionospheric activity is high. Finally, the *intermediate* data is during periods of lower ionospheric activity (relative to early 2015) but before the GPS receiver updates. In total we have selected 7 test months: January and March 2015 refer to *bad* data quality; February and March 2016 refer to *intermediate* data quality; and June-August 2016 refer to *good* data quality.





**Table 3.** Periods considered in the analysis of the added value of different types of non-gravitational accelerations

| Period | Accelerometer artefact density | Ionospheric activity | Geomagnetic activity | Accelerometer signal magnitude |
|---|---|---|---|---|
| January 2015 | high | high | low | high |
| February 2015 | middle | middle | low | high |
| March 2015 | low | high | high | high |
| January 2016 | middle | low | low | low |
| February 2016 | middle | low | low | low |
| March 2016 | low | low | low | low |

The existing software exploiting VEA at IfG handles the Swarm KB data under the same processing scheme and handling
of stochastic properties of the observations adopted for the generation of the ITSG-GRACE releases (Mayer-Gürr et al., 2016).
The observations derived from the Swarm KBs are introduced into the gravity inversion process as if they were collected by
the K-Band ranging instrument. Our software is not prepared to handle the full three-dimensional (3D) information of the KBs
and the development of this capability is outside the scope of this study.

The KBs and KO solution are selected consistently from the same AC (i.e. TUD or AIUB) when producing the gravity
field solution. In total 4 different GFM variants have been computed: (1) hl-SST solution from TUD KOs, (2) hl-SST+low-
low Satellite-to-Satellite Tracking (ll-SST) solution from TUD KOs and KBs, (3) hl-SST solution from AIUB KOs, and (4)
hl-SST+ll-SST solution from AIUB KOs and KBs. The four solution variants were produced for all seven test months.

### 2.3 Non-gravitational accelerations

We assessed the quality quality of the Swarm GFMs when the non-gravitational accelerations are modelled following two
distinct approaches and when they are represented by the L1B accelerometer data from Swarm-C (Siemes et al., 2016). One
non-gravitational acceleration model was produced at Astronomical Institute Ondřejov (ASU) and the other at Delft University
of Technology (TUD). We selected a number of periods for our tests (cf. Table 3), taking care to cover as much as possible
different accelerometer data variability (arising from instrument artefacts) and signal amplitude, as well as ionosphere and
geomagnetic activity, to cover different regimes of non-gravitational accelerations acting on the Swarm satellites. Moreover,
we also chose months when GRACE gravity field solutions are available, to facilitate validating the Swarm GFMs.

For the ASU model, we used the in-house orbital propagator NUMINTSAT (Bezděk et al., 2009) for processing the satel-
lite orbital data, computing the coordinate transformations and generating the modelled non-gravitational accelerations of each
Swarm satellite. The computation of the non-gravitational acceleration forces requires the knowledge of the physical properties



of the satellite based on the information provided by ESA: its mass, cross-section in a specific direction, radiation properties

of the satellite's surface and a macro model characterizing approximately the shape of the Swarm satellites. For neutral atmospheric density, we made use of the  Naval Research Laboratory Mass Spectrometer and Incoherent Scatter Radar (NRLMSISE) model (Picone et al., 2002). We estimated the drag coefficient of each satellite by means of the long-term change in the orbital elements in order to consider realistic values. Further details of our approach can be found in Bezděk (2010); Bezděk et al. (2014, 2016, 2017).

For the TUD model, the Near Real-Time Density Model (NRTDM) software was employed (Doornbos et al., 2014). This software, as part of the "official" Swarm Level 2 Processing System (L2PS) infrastructure, is used in the L1B to Level 2 (L2) processing at TUD. A variety of models and parameters related to the non-gravitational forces is available in this software. For the current study, the following selection was made: the Swarm panel model (macro model) is based on (Siemes, 2019); the panel orientation is dictated by Swarm quaternion data; the satellite aerodynamics of single-sided flat panels are computed

following Sentman's equations (Sentman, 1961), assuming diffuse reflection and energy flux accommodation set at 0.93; the neutral densities are derived from the NRLMSISE thermosphere model, as well as temperature and composition-dependence of Sentman's equations; the velocity of the atmosphere with respect to the spacecraft is based on the orbit and attitude data, atmospheric co-rotation and modelled thermospheric wind using the Horizontal Wind Model 07 (HWM07) (Drob et al., 2008) and the Disturbance Wind Model 07 (DWM07) (Emmert et al., 2008); the Solar Radiation Pressure (SRP) is computed taking

into account absorption, diffuse reflection and specular reflection, according to optical properties of the surface materials supplied by ESA and Astrium, and it considers the varying Sun-satellite distance; the Sun-Earth eclipse model takes into account atmospheric absorption and refraction, according to the Analysis of Non-Gravitational Accelerations due to Radiation pressure and Aerodynamics (ANGARA) implementation (Fritsche et al., 1998); the Earth Infrared Radiation Pressure (EIRP) and Earth Albedo Radiation Pressure (EARP) are based on the ANGARA implementation, and monthly average albedo coefficients and

Infrared Radiation (IR) irradiances from Earth Radiation Budget Experiment (ERBE) data (Barkstrom and Smith, 1986). The equations for the algorithms and references for these models are available in Doornbos (2012) with updates specific to Swarm provided by Siemes et al. (2016).

For the Swarm-C accelerometer data, we took advantage of the corrected L1B along-track ACC data (Siemes et al., 2016), which is distributed by ESA and processed in a single batch from July 2014 to April 2016. We applied a dedicated calibration

method to the Level 1A (L1A) product `ACCxSCI_1A` for the cross-track and radial components (Bezděk et al., 2017, 2018b) but, as shown by Bezděk et al. (2018a), this approach was unable to recover the expected signal. For this reason, the non-gravitational acceleration measurements are restricted to the available along-track Swarm-C data.

## 2.4   Gravity Field Models estimation approaches

The estimation of the hl-SST GFMs takes the KOs as observations, which describe the satellite's Centre of Mass (CoM)

motion since in their production, the processing of the L1B GPS measurements is corrected for location of the GPS antenna phase centre with the L1B Swarm attitude data. The KOs are suitable to gravimetric studies due to their purely geometric nature. Through a parameter estimation procedure, i.e. one of the strategies listed in Table 4, the gravity field parameters are



derived from a functional relationship between the kinematic positions and gravity field parameters. Complementary to the KOs, numerous processing choices are made by the four gravity field ACs, as enumerated in Appendix C

Each AC selects one KO solution to produce their so-called *individual* GFMs, as listed in Table 4. In contrast, the *combined* GFMs are derived from these individual solutions, as discussed in Section 2.5. The following subsections provide a brief recap of the selected methods. Elaborate details can be found in the referenced literature.

**Table 4.** Overview of the gravity field estimation approaches

| Inst. | Approach | Reference |
|---|---|---|
| AIUB | Celestial Mechanics Approach (Beutler et al., 2010) | Jäggi et al. (2016) |
| ASU | Decorrelated Acceleration Approach (Bezděk et al., 2014, 2016) | Bezděk et al. (2016) |
| IfG | Short-Arcs Approach (Mayer-Gürr, 2006) | Zehentner and Mayer-Gürr (2016) |
| OSU | Improved Energy Balance Approach (Shang et al., 2015) | Guo et al. (2015) |

### 2.4.1    Celestial Mechanics Approach

The Celestial Mechanics Approach (Beutler et al., 2010), used at AIUB, is a variation of the traditional variational equations
approach (Reigber, 1989), which linearises the relation between the kinematic positions and the unknown Stokes coefficients as well as other unknown parameters that play a role in the dynamic model described by the equations of motion, such as initial state vectors, empirical accelerations, drag coefficients, instrument calibration parameters, (possibly) amongst others. Pseudo-stochastic pulses or accelerations are estimated to mitigate deficiencies of the a priori force model. The CMA has successfully been applied for gravity field determination from a number of LEO satellites, e.g. Meyer et al. (2019b).

### 2.4.2    Decorrelated Acceleration Approach

The Decorrelated Acceleration Approach (DAA) (Bezděk et al., 2014, 2016), used at ASU connects the double-differentiated kinematic positions to the external forces acting on the satellite. This approach computes the geopotential harmonic coefficients from a linear (not linearised) system of equations. The observations are first transformed to the inertial reference frame before differentiation to avoid the computation of fictitious accelerations. The differentiation of noisy observations leads to the
amplification of the high-frequency noise. However, it is possible to mitigate the high-frequency noise with a decorrelation procedure. We apply a second decorrelation based on a fitted autoregressive process to take into account the error correlations of the KOs.





### 2.4.3 Improved Energy Balance Approach

The traditional Energy Balance Approach (EBA) exploits the energy conservation principle to build a relation between the
residual geopotential coefficients (relative to the reference background force model) and the deviations of the KO from the ref-
erence orbit on (Jekeli, 1999; Visser et al., 2003; Guo et al., 2015; Zeng et al., 2015). The main development of the IEBA, used
at Ohio State University (OSU), concerns the handling of the noise in the kinematic position and the weighting of the potential
observations. Unlike the application of this approach to GRACE ll-SST data by Shang et al. (2015), the term related to the
Earth's rotation cannot be neglected in the processing of hl-SST data. From the kinematic positions, the velocity is derived with
a 61 data points, sliding window, quadratic polynomial filter similar to Bezděk et al. (2014). The polynomial coefficients of
the filter are estimated in a LS adjustment, with the observation vector being composed of position residuals between the kine-
matic positions and the corresponding reduced-dynamic positions (integrated on the basis of the reference background force
model), and the observation covariance matrix constructed from the epoch-wise variance-covariance information distributed
in the KOs data files. As a consequence of this orbit smoothing procedure, we discard the warm-up/cool-down edges of the
daily data arcs. We further remove 1 Cycle Per Revolution (CPR) sinusoidal and 3-hourly quadratic polynomial signals from
the potential observations derived from the smooth kinematic positions. We also take advantage of the observation covariance
matrix to weight the filtered kinematic observations in the geo-potential coefficient LS inversion. We do not apply any a priory
constraints nor iterate the LS estimation since we take advantage of the linear relation between the potential observations and
the geo-potential coefficients.

### 2.4.4 Short-Arcs Approach

The Short-Arcs Approach (Mayer-Gürr, 2006), used at IfG, formulates the relation between the geopotential coefficients and
the kinematic positions as the boundary value problem resulting from the double-integration of the equations of motion. This
approach naturally defines the initial state vector as the boundary conditions of the integral equation, which are regarded as
unknowns in the LS estimation along with the Stokes coefficients and other unknown parameters, such as empirical parameters.
Additionally, the kinematic positions are treated with no explicit differentiation, thus circumventing the need to suppress the
amplification of high-frequency noise.

### 2.5 Combination

The individual GFMs are combined in the frame of the Combination Service of Time-variable Gravity Fields (COST-G) of
the IGFS, applying the methods developed in the frame of the EGSIEM (Jäggi et al., 2019). We derive VCE weights in order
to produce the combined GFMs from the individual GFMs produced at AIUB, ASU, IfG and OSU. The VCE weights are
derived on the solution level according to Jean et al. (2018), considering the individual models up to degree 20 only; if this
is not done, the extremely high noise at the degrees close to 40 (the maximum degree of the individual solutions) dominates
the estimation of the weights, which leads to a slightly worse agreement with GRACE (Teixeira da Encarnação and Visser,
2019). Irrespective of this, the maximum degree of the combined models is the same as the individual models (degree 40).





We also tested the combination at the level of Normal Equations (NEQs) (Meyer et al., 2019a) but determined that the signal
content was not in as good agreement with GRACE as the combination at the level of solutions with weights derived from VCE
(Teixeira da Encarnação and Visser, 2019; Meyer, 2019). We attributed this result to the difficulty in calibrating the formal error
types resulting from the different gravity field estimation techniques. There is the issue of the different types of error: some
provide calibrated errors (e.g. DAA), while others provide the formal errors from the LS estimates (e.g. CMA). Another issue
is the different error amplitude dependence with degree, thus preventing the errors to be calibrated with a simple bias. Finally,
the time-dependent levels of errors in the individual models, which change their fidelity with time, and consequentially their
optimum relative weights, were also a factor preventing us from successfully performing a combination at the NEQ level.

## 2.6    Assumptions in the Gravity Field Model analyses

This section describes the set of assumptions considered in the analysis done in Sections 3.3 and 3.4. Sections 3.1 and 3.2
report parallel studies that were conducted with different background force models, better suited to their respective purposes.

We have chosen the Release 6 (RL06) GRACE and GRACE-FO GFMs produced at Center for Space Research (CSR) as
comparison in our analysis of the Swarm GFMs. At the spatial scales relevant to Swarm, we have no reason to expect our
results would change significantly if GRACE data produced at any other AC was used instead.

Unless otherwise noted, we apply a $750\mathrm{km}$ radius Gaussian smoothing, which we motivate in Section 3.4.1, to isolate the
signal content in the Swarm models. The geo-centre motion has been ignored in our analysis, i.e. the degree 1 coefficients are
always zero. The Combined GRACE Gravity Model 05 (GGM05C) static GFM (Ries et al., 2016) is subtracted from all Swarm
and GRACE solutions in order to isolate the time-variable component of Earth's gravity field. The gravity field is presented
in terms of Equivalent Water Height (EWH), except for the statistics related to the correlation coefficient or when presenting
coefficient-wise time series.

We consider the entirety of the Swarm GFM time series, irrespective of the epoch-wise quality because our objective is
to give a complete overview of the quality and characteristics of our models. The analysis spans all available months during
the Swarm mission, i.e. between December 2013 and March 2019. The GRACE time series is linearly interpolated to the
time domain defined by the mid-month epoch of the Swarm solutions, except for the GRACE/GRACE-FO gap, where no
interpolation is performed.

### 2.6.1    Earth's oblateness

In our analysis, the proper handling of Earth's oblateness is not a trivial problem. In case of GRACE, the mass estimates are
improved if $C_{2,0}$ is augmented with Satellite Laser Ranging (SLR) data, which are provided in the form of the time series
produced by Cheng and Ries (2018). Therefore, any comparison with mass variations derived from Swarm must also have the
$C_{2,0}$ coefficient replaced by the same time series. One could argue that simply discarding this coefficient would suffice for
any comparison but we also intend to represent the actual mass changes observed by Swarm, notably in Section 3.5.3, where
we show mass variations over large storage basins. Unfortunately, Earth's oblateness estimates provided by Cheng and Ries





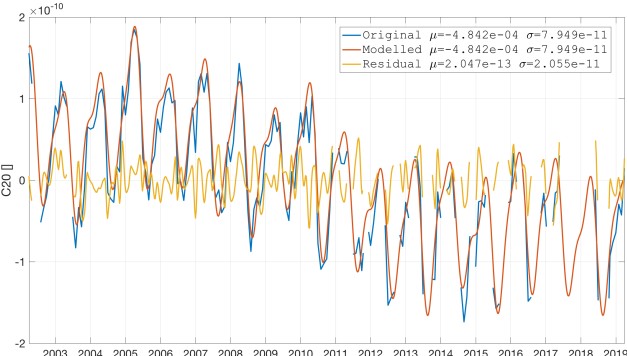

**Figure 1.** Earth oblateness model (*Modelled*) resulting from first order polynomial coefficients and 10 sinusoidal periods, determined in a data-driven approach; time series of GRACE and SLR estimates of the $C_{2,0}$ coefficient (*Original*); difference between these two time series (*Residual*). The legend reports the mean $\mu$ of all time series (removed before plotting) as well as their STD $\sigma$.

(2018) are exclusively available at those epochs when there are GRACE solutions. That essentially means that interpolating these GRACE/SLR $C_{2,0}$ estimates over large gaps would lead to unrealistic mass variations.

We circumvent this problem by representing the Cheng and Ries (2018) time series with a dedicated Earth oblateness temporal model composed of polynomial coefficients and a series of sinusoidal periods. Unlike the GRACE climatological model (presented in Section 2.6.3), there are no clear candidates for the periods that compose this Earth oblateness temporal model. To address this problem, we implemented a data-driven procedure that iteratively finds characteristic periods in the data. We begin with a parametric model that includes first order polynomial coefficients, as well as yearly and semi-yearly sinusoidal periods. The residual between the original data in Cheng and Ries (2018) and the model output undergoes a Fourier analysis to determine the period with highest amplitude. In the following iteration, this period is included in the parametric model and the subsequent residual is again subjected to the Fourier analysis to determine the new period with highest amplitude. The procedure is repeated until no new periods are found.

Figure 1 demonstrates that a first order polynomial and the 10 sinusoidal periods listed in Table 5 (to a total of 22 parameters) are able to represent 75% of the variability in the Cheng and Ries (2018) time series, since the STD of the residual time series is 25% of the original one.

### 2.6.2 Deep ocean areas

We consider the ocean mask of the areas away from continental masses illustrated in Figure 2. To produce this mask, we start with a a grid with unit value over land areas, convert it to the Spherical Harmonic (SH) domain, apply Gaussian smoothing with a radius of 1000km, convert it back to the spatial domain and define those grid points with values below the cut-off value of 0.9 to be in deep ocean areas. The cut-off value was selected on the basis of trial and error with the objective of generating an ocean mask with the desired and arbitrary buffer length, which for the results reported here remained equal to 1000km.



**Table 5.** Coefficients of the continuous Earth oblateness model derived from Cheng and Ries (2018), with zero epoch at 2002/4/1 00:00:00.

| Poly. Coeff. | Value | |
|---|---|---|
| bias [ ] | $-0.00048416$ | |
| trend [/year] | $-3.27564 \times 10^{-14}$ | |
| Period [years] | Sine [ ] | Co-sine [ ] |
| 1 | $-2.85714 \times 10^{-11}$ | $6.71883 \times 10^{-11}$ |
| 0.5 | $2.27596 \times 10^{-11}$ | $8.87782 \times 10^{-12}$ |
| 10.51 | $-3.60958 \times 10^{-12}$ | $-2.58964 \times 10^{-11}$ |
| 5.26 | $-9.56044 \times 10^{-12}$ | $-5.53426 \times 10^{-12}$ |
| 2.34 | $6.88445 \times 10^{-12}$ | $2.97893 \times 10^{-12}$ |
| 3.01 | $-7.66733 \times 10^{-12}$ | $-7.74553 \times 10^{-14}$ |
| 1.50 | $6.09057 \times 10^{-12}$ | $6.76943 \times 10^{-12}$ |
| 4.21 | $-1.08007 \times 10^{-11}$ | $3.19040 \times 10^{-12}$ |
| 0.73 | $8.16837 \times 10^{-12}$ | $6.58205 \times 10^{-12}$ |
| 0.64 | $-1.72176 \times 10^{-12}$ | $-2.16154 \times 10^{-12}$ |

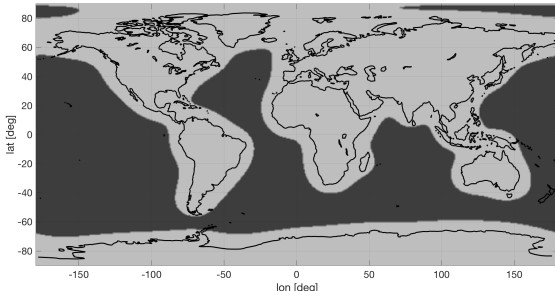

**Figure 2.** Deep ocean mask shown as dark areas.

This procedure pushes the boundary of an ocean mask away from continental coastal areas and ignores islands. For the spatial scales relevant to the Swarm GFMs, we propose that this procedure is adequate.

### 2.6.3 GRACE climatological model

In the analyses conducted in later Sections, we use a parametric representation of Earth's temporal mass changes as observed by GRACE, which we refer to as *climatological model* since it captures mass variations that are present in all 15 years of GRACE data. We do not use any GRACE-FO data in this regression, in order to be able to verify the continuity of the GRACE-FO data, relative to GRACE and if substantiate any deviation that is also observed by Swarm. This parametric regression is performed on the original CSR RL06 models, i.e. before any smoothing or masking.





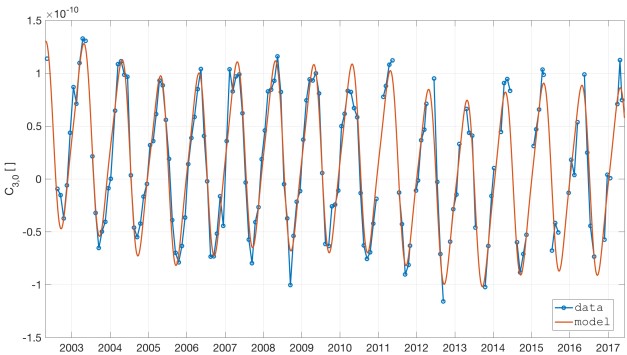

**Figure 3.** Agreement between the GRACE climatological model and the GRACE data, exemplified by the $C_{3,0}$ coefficient.

We selected the first order polynomial to represent bias and trend in the GRACE data. For the periodic parameters, we choose the year and semi-year periods since these are dominant signals in the GRACE and Swarm data. We also modelled the S2, K2 and K1 tidal periods, with durations of $0.44$, $3.83$ and $7.67$ years, respectively. These periods drive the orbital inclination of the GRACE satellites and produce strong aliasing in the GFM time series (Ray and Luthcke, 2006; Cheng and Ries, 2017). The linear regression of the 12 parameters is done independently for each SH coefficient, up to degree 40 (in agreement with

the maximum degree of the Swarm models). This results in 12 sets of Stokes coefficients, one for each of the model parameter: bias, trend and 5 periods represented by their sine and co-sine components. Each set of parametric Stokes coefficients has an implicit time dependence which is evaluated coefficient-wise at the epochs of the Swarm GFMs. We illustrate the general agreement between the climatological model and the GRACE data for the case of $C_{3,0}$ in Figure 3.

We regard this model as good representation of the Earth system; it is by definition inferior to the original GRACE time series

because it truncates the signal bandwidth to discrete frequencies. In spite of this, the assumed climatological model provides a measure to which both GRACE and Swarm can be compared. The differences between GRACE and this model should be regarded as the signal augmentation that GRACE brings, not as an error. As we illustrate later, the signal augmentation is of substantial lower amplitude than the model. Therefore, the Swarm residual with respect to the climatological model can be safely regarded as errors, unless of amplitude comparable with the GRACE residual (which is never the case). We also

regard the vastly different spatial sensitivity of Swarm compared to GRACE as an additional argument that the climatological model is able to represent the Earth system in a much more accurate way than Swarm, with the exception of large atypical mass variations (which are uniquely revealed by Swarm). In this sense, we regard the climatological model as *good enough* to quantify the quality of the Swarm

## 3    Results

Our results are shown in the following Section, where we analyse the added value of KBs in Section 3.1, look into the effect of including accelerometer measurements of Swarm-C in Section 3.2, provide an overview of the quality of our individual



solutions in Section 3.3, quantify the quality of the combined solutions in Section 3.4 and illustrate their signal content in Section 3.5.

## 3.1 Kinematic Baselines

This section is dedicated to quantifying the benefit of exploiting KBs in the quality of the gravity field models derived from Swarm data, following the motivation and procedures described in Section 2.2.

Due to the decreasing ionospheric activity and the changes made to the Swarm on-board GPS receivers between 2015 and 2016 (van den IJssel et al., 2016), the consistency of the KB solutions has improved. Especially in summer 2016, the overall daily STD of the difference between the reduced dynamic ambiguity-fixed and kinematic ambiguity-fixed baselines may be as 355 low as $10-15$mm, $4-6$mm and $3-5$mm on average for the radial, along-track and cross-track directions, respectively, while it is as high as $1-3$cm for 2015 in all 3 directions. It should be noted, however, that daily STD is always dominated by the low quality kinematic positions over the polar regions. Eliminating such problematic data, the difference STD is consistently under $5$mm; therefore, the internal precision of the Swarm GPS data is of very good quality.

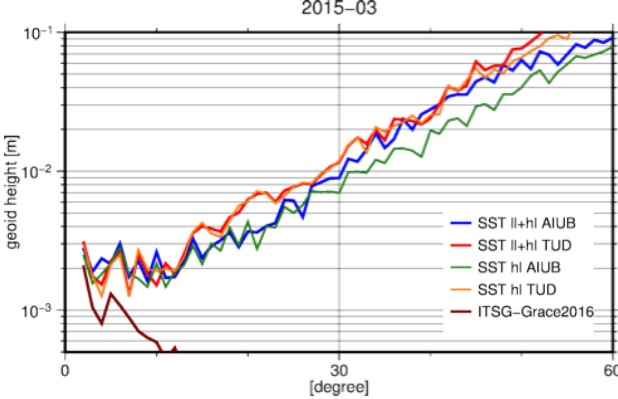

**Figure 4.** Difference SH degree amplitudes of all four test solutions for March 2015 (regarded as representative of *bad* data quality) with respect to GOCO05S.

Figures 4 to 6 show the difference degree amplitudes with respect to the static part of the GOCO release 05 satellite-360 only gravity field model (GOCO05S) (Mayer-Gürr, 2015) in terms of geoid heights, representative of the results for *bad*, *intermediate* and *good* data quality. For comparison the corresponding month from the ITSG GRACE-only model, 2016 (ITSG-GRACE2016) time series is also shown. For all months it can be seen that the solutions do not differ significantly. There are small differences between the two ACs (AIUB and TUD) as well as between the hl-SST-only and the ll-SST+hl-SST solutions. Differences are larger for those months with "bad" data quality (2015) and at the SH degree regions dominated by 365 noise (above degree 15), with the ll-SST/hl-SST solutions showing larger degree amplitudes. For months with "good" data quality (June 2016) all four solutions display much smaller differences.

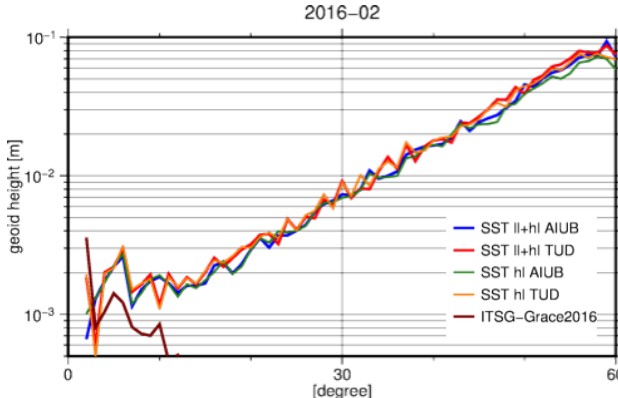

**Figure 5.** Difference SH degree amplitudes of all four test solutions for February 2016 (regarded as representative of *intermediate* data quality) with respect to GOCO05S.

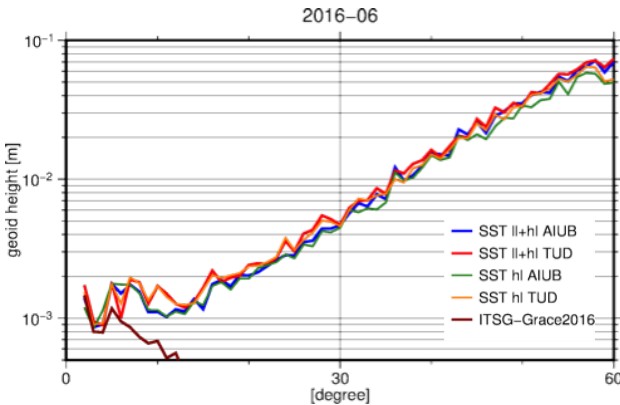

**Figure 6.** Difference SH degree amplitudes of all four test solutions for June 2016 (regarded as representative of *good* data quality) with respect to GOCO05S.

To quantify the impact on the long wavelength part of the solutions, we have compared the individual solutions to ITSG-GRACE2016 monthly solutions in spatial domain. The solutions are evaluated on a equiangular grid ($1° \times 1°$), reduced by the corresponding ITSG-GRACE2016 monthly solution, filtered with a 500km Gaussian filter, and finally the RMS over all grid cells is com-

puted. The filter width was selected so as to avoid suppressing all of the signal at degrees above 20 in order to assess the impact of KBs on the high-frequency noise as well. These results are summarized in Table 6.

Table 6 confirms what is depicted in Figures 4 to 6, i.e. the inclusion of KBs in the gravity field estimation has no significant impact on the quality of the resulting GFM. KO-only solutions are already of very similar quality when compared to KB-augmented solutions, with small differences visible in the degree amplitudes plots or the spatial RMS having no discernible

correlation with the data period (and therefore, quality). In general, this confirms the findings of Jäggi et al. (2009), in that there are some small benefits for higher degrees when using KB; this was attributed to the elimination of errors common to





**Table 6.** RMS of geoid height differences in mm for different hl-SST-only and the ll-SST/hl-SST Swarm solutions with respect to the corresponding ITSG-GRACE2016 monthly solution.

| Data quality | Solution | TUD | | AIUB | |
|---|---|---|---|---|---|
| | | hl-SST | ll-SST+hl-SST | hl-SST | ll-SST+hl-SST |
| *bad* | January 2015 | 9.5 | 9.6 | 9.8 | 10.5 |
| | March 2015 | 10.9 | 11.1 | 8.4 | 9.6 |
| *intermediate* | February 2016 | 7.5 | 7.4 | 7.4 | 7.2 |
| | March 2016 | 8.8 | 8.6 | 7.3 | 7.3 |
| *good* | June 2016 | 5.4 | 5.5 | 4.8 | 4.8 |
| | July 2016 | 6.7 | 6.5 | 6.3 | 6.1 |
| | Aug. 2016 | 5.7 | 5.8 | 5.3 | 5.4 |

both satellites by using DD observations. Our results suggest that common errors are already mostly absent in the computation of the Swarm KOs. Thus we found no added value in including KBs to the quality of Swarm GFMs.

Our results contrast with Guo and Zhao (2019), who demonstrated a noticeable improvement when KBs are used in conjunction with KOs to derive gravity field models from hl-SST GRACE data. As the authors mention, their approach benefits from the 3D KB information, thus essentially increasing by a factor of 3 the number of observations. Although these components are most likely not completely independent, they provide observations with crucial information that is not available along the Line of Sight (LoS) component, in particular along the radial direction. We also note that the geometry of the GRACE formation provides a much more stable amplitude and attitude of resulting KBs, which may benefit the ambiguity fixing and, consequently, their overall quality. In case of Swarm, the KBs are close to zero and flip their orientation by $180°$ at the poles. Additionally, GRACE accelerometer data were used to represent the non-gravitational accelerations, which less straightforward for the Swarm satellites. These differences, i.e. 3D baselines, stable baseline length and inclusion of accelerometer data, suggest that they may be necessary conditions for a positive added value of KBs to the quality of hl-SST-only gravity field models. Finally, we also point out that the improvements reported in Guo and Zhao (2019) are only above SH degree 10, where the errors start to become dominant, thus reducing the practical added value of including baselines in the estimation of hl-SST-only GFMs.

### 3.2 Non-gravitational accelerations

In this section, we present the inter-comparison of the three types of non-gravitational accelerations described in Section 2.3. Figure 7 compares three single-satellite gravity field solutions derived from Swarm-C data, considering the three non-gravitational accelerations, for January 2015.

The SH degree difference amplitudes illustrate that the measured non-gravitational accelerations improve the agreement of the lowest degrees of the Swarm-C monthly solution with respect to the GOCO05S model (Mayer-Gürr, 2015), which includes



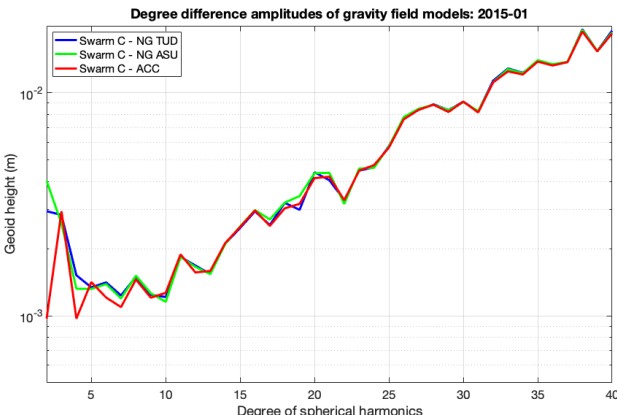

**Figure 7.** Swarm-C gravity field solutions using TUD and ASU modelled non-gravitational accelerations, as well as measured non-gravitational accelerations (January 2015).

a time-variable component. We tested this comparison relative to the ITSG-GRACE2016 monthly GFM (Mayer-Gürr et al., 2016) and observed similar results (not shown). The improvement at the lowest degrees in the Swarm-C model when using observed non-gravitational acceleration data is in accordance with what was reported by e.g. Klinger and Mayer-Gürr (2016), relative to GRACE gravity field recovery.

In view of the lack of reliable measured non-gravitational accelerations in Swarm-A and Swarm-B, the three-satellite Swarm GFM considers the ASU modelled non-gravitational accelerations for these satellites. For Swarm-C, we consider three cases where the non-gravitational accelerations are either measured or represented by TUD or ASU's model. In this way, we isolate the effect of the three types of non-gravitational acceleration data. The results for January 2015 are shown in Figure 8, using ASU and TUD models, and calibrated accelerometer data.

The three-satellite solutions that use modelled non-gravitational accelerations in Swarm-C are remarkably similar (cf. Figure 8). In spite of this, note that using accelerometer data improved the agreement to GOCO05S for degrees 2 and 4.

To gather a better overview of the added value of the three types of non-gravitational accelerations, we derive the following model difference $D$, similar to RMS:

$$D = \sqrt{\text{median}(\Delta h)^2 + \text{MAD}(|\Delta h|)^2} \tag{1}$$

with the Median Absolute Deviation (MAD) an analogous to STD when the median is considered instead of the mean and $\Delta h$ being the $1° \times 1°$ geoid height difference between the $500\text{km}$ Gaussian filtering three-satellite Swarm models and both ITSG-GRACE2016 and GOCO05S, in the latitude band $85°$ from the equator. We note that similar results were obtained using the CSR RL05 GRACE monthly solutions (not shown). The resulting differences are shown in Table 7.

The 2015 results indicate that the observed non-gravitational accelerations improve the agreement between the three-satellite Swarm models and ITSG-GRACE2016/GOCO05S, while that is not the case for 2016 (except for January 2016 and GOCO05S,





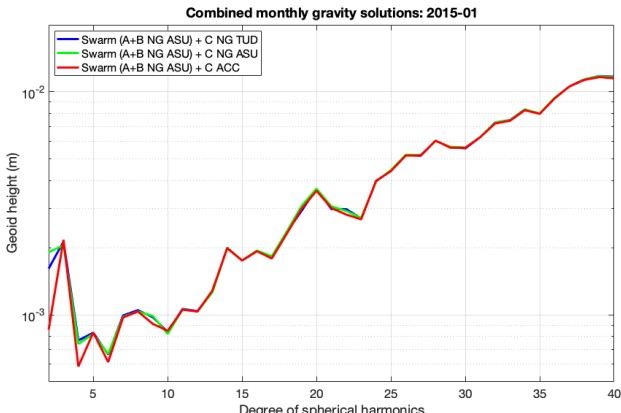

**Figure 8.** Three-satellite Swarm gravity field solutions using TUD and ASU modelled non-gravitational accelerations, as well as measured non-gravitational accelerations for Swarm-C and ASU modelled non-gravitational accelerations for Swarm-A and B (January 2015).

**Table 7.** Geoid height difference in mm between Swarm and GRACE GFMs.

|  | ITSG-GRACE2016 | | | GOCO05S | | |
|  | mod. ASU | mod. TUD | obs. | mod. ASU | mod. TUD | obs. |
|---|---|---|---|---|---|---|
| January 2015 | 16.2 | 15.6 | 15.0 | 16.7 | 16.5 | 15.9 |
| February 2015 | 18.8 | 18.0 | 17.9 | 18.0 | 17.7 | 17.5 |
| March 2015 | 16.4 | 16.5 | 16.1 | 16.2 | 16.3 | 16.0 |
| January 2016 | 20.3 | 20.0 | 20.5 | 17.5 | 17.3 | 17.3 |
| February 2016 | 23.9 | 22.3 | 25.6 | 15.2 | 14.3 | 16.3 |
| March 2016 | 17.1 | 15.6 | 18.5 | 12.5 | 12.4 | 12.9 |

when the GFM derived from TUD modelled non-gravitational accelerations agree equally well with the one derived considering observed non-gravitational accelerations). The comparison with GOCO05S intends to predict how well would it be possible to assess the added value of the different types of non-gravitational accelerations during those periods when there are no GRACE data. Other time-dependent models were tested but those do not agree as closely with GRACE monthly models (not shown).

The statistics in Table 7 imply that observed non-gravitational accelerations are only beneficial when the amplitude of the non-gravitational accelerations is larger than what was observed in 2016. This is likely related to the decreasing level of solar activity, which is approaching the minimum of its 11-year cycle (expected to reach the minimum in 2019). Through the influence of the solar radiation on the atmospheric density and resulting atmospheric drag, the low level of solar activity has a direct impact on the accelerometer measurements. The closer to the solar cycle minimum, the lower magnitude and variability of the accelerometer signal is. Another factor may be a potential worse performance of the accelerometer calibration procedure under low levels of solar activity, resulting from the lower Signal-to-Noise Ratio (SNR) in the accelerometer data. In other

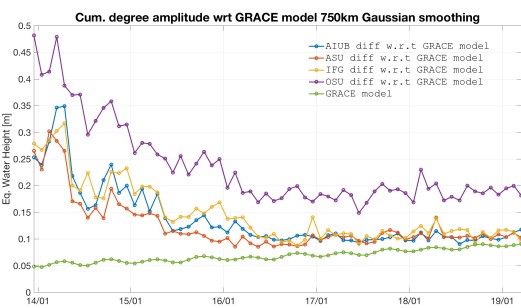

**Figure 9.** Agreement between the various individual Swarm monthly solutions and the GRACE climatological model in terms of cumulative degree RMS (maximum degree 20) as function of time, considering 750km smoothing.

words, the noise and (potentially) uncorrected artefacts in the accelerometer data of Swarm-C are substantial enough to limit the usefulness of these data to gravimetric studies, except when the solar activity is high (as was the case in 2015) or when

the satellites' altitude decays in the future. Given these characteristics and the continuing solar minimum, our Swarm models are not processed considering Swarm-C accelerometer observations, but we plan to revisit this issue once the solar activity increases.

### 3.3 Individual Swarm models

In this section we illustrate the quality of the individual Swarm solutions. As described in Section 2.6, we consider a GRACE

climatological model defined by 12 parameters as a "good enough" representation of the Earth system at the spatial lengths observed by Swarm, i.e. under 750km radius Gaussian smoothing.

Figure 9 shows a measure of the evolution of the quality of the individual Swarm solutions over the complete Swarm data period. We also plot the cumulative degree amplitude of the GRACE climatological signal, to illustrate the global spatial amplitude of the geophysical processes represented by this model. There is a clear improvement in the agreement of Swarm

with GRACE, from RMS differences as high as 50cm geoid height in early 2014, down to 10cm since 2016. We attribute this increase in quality to the decrease in solar activity and to the upgrades in the Swarm GPS receivers between 2015 and 2016 (van den IJssel et al., 2016; Dahle et al., 2017). As demonstrated in Section 3.4, the Swarm models contain large errors in the ocean areas, which dominate the global spatial RMS difference; over land areas, the agreement with GRACE is much better.

The various individual solutions show different levels of quality. Generally speaking, the solutions from AIUB, ASU and

IfG cluster together as agreeing better with GRACE, with their dispersion narrowing down after 2016. This suggests that these approaches suffer differently in conditions of high solar activity, with ASU's models being the least sensitive overall. Possibly, ASU's efforts to minimize the amplification of the high frequencies when performing the double differentiation of the kinematic positions has the side effect of suppressing the negative effects of the high solar activity in the quality of the kinematic orbits. In contrast, OSU's solution consistently has lower agreement with GRACE. The velocity measurements, which are needed for

IEBA (as well as any EBA-type approach), are to be derived from the kinematic positions by differentiation (then squared to

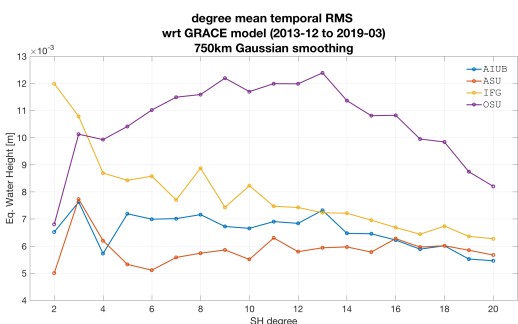

**Figure 10.** Agreement between the various individual Swarm monthly solutions and the GRACE climatological model in terms of the degree mean of the per-coefficient temporal RMS difference, considering 750km smoothing.

obtain kinetic energy). The tedious data filtering and processing to approximate velocity errors is still imperfect, particularly in light of the spurious jumps in most of the kinematic orbits even in the cases without the GPS tracking signal degradation, e.g. from the Southern Atlantic anomaly.

Another way of analysing the agreement between the individual solutions and GRACE is to derive per-coefficient statistics of their temporal variations. One such statistic is the coefficient-wise temporal RMS of the difference between the Swarm individual solutions and the climatological model, thus producing a set of Stokes coefficients that describes the variability of that difference; from this set we compute the mean over each degree to represent the general agreement at the corresponding spatial wavelengths. The results are summarized in Figure 10, which quantifies the agreement of Swarm and the GRACE climatological model in the spectral domain. Note that for most individual solutions, the RMS difference decreases with degree as result of the Gaussian smoothing, without which the curves would have a strong overall positive slope.

The ranking of quality of the individual solutions changes with spatial wavelength; for example, although OSU's solutions are consistently worse than IfG's as shown in Figure 9, their degrees 2 and 3 are on average in better agreement with GRACE. This diversity in the particularities of the various solutions is the main motivation for our practice to combine solutions derived from multiple gravity field estimation approaches. Unfortunately, as explained in Section 2.5, our combination is done at the solution level with weights derived from VCE, which means we loose the ability to weight the individual solutions differently in the degree domain and we cannot fully take advantage of the per-degree variations in quality of the individual solutions. Nevertheless, the VCE weights produce combined solutions with better agreement to GRACE than those combined at the NEQ level (Teixeira da Encarnação and Visser, 2019). From this we interpret that the benefits from per-degree weighting may not be as significant as the disadvantages of the combination at NEQ level, namely the different types of formal/calibrated errors, their different temporal evolution and the difficulty in finding adequate empirical weights.

Finally, Figure 11 shows the correlation of the Swarm time series with GRACE for the relevant spatial wavelengths. This figure is complementary to Figure 10, since it does not illustrate the overall agreement (which is a measure of error) but the level that Swarm observes the same temporal evolution as GRACE (i.e. if Swarm sees the same proportional mass increase/decrease as GRACE). Understandably, the highest correlations correspond to the lowest degrees, not only because those are the signals





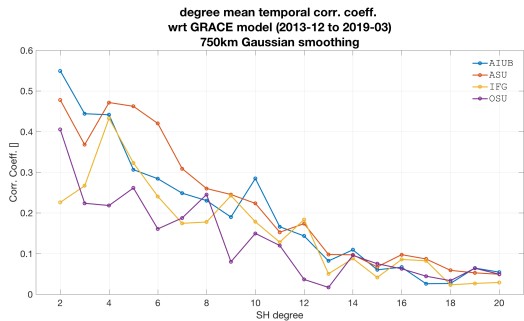

**Figure 11.** Agreement between the various individual Swarm monthly solutions and the GRACE climatological model in terms of the degree mean of the per-coefficient temporal correlation, considering 750km smoothing.

with highest amplitude (and therefore better observed by Swarm and GRACE) but also because of the smoothing. There is no obvious individual solution that stands out as being better correlated with GRACE, although ASU has the highest correlation coefficient for degrees 4 to 8, while for AIUB that is the case for degrees 2 and 3. OSU's solution tends to correlate the least, except for degrees 2, 7 and 8; this again indicates that a solution that may at first seem to be of consistently inferior quality may still provide a positive contribution to the combination. Also note that the correlations drop below 0.1 above degree 12

and remain relatively constant for higher degrees, indicating there is very little signal in the individual solutions that represent the same temporal variations as GRACE.

### 3.4   Combined Swarm models

Having presented the individual Swarm GFMs in the previous section, we dedicate the current section to the analysis of the combined solutions. For more details about the combination strategy, refer to Teixeira da Encarnação and Visser (2019) and

Meyer (2019). We determine the necessary intensity of smoothing of the Swarm models (Section 3.4.1) and illustrate the different sensitivity of the Swarm data to observe mass transport processes over land and ocean areas (Section 3.4.2).

#### 3.4.1   Smoothing of the Swarm solutions

As demonstrated by Teixeira da Encarnação et al. (2016), the Swarm models do not seem to be sensitive to full wavelengths shorter than roughly 1500km. We now update this assessment in light of the much longer time series and improved combination

strategy than was the case in earlier publications. We compute the cumulative degree amplitude (which is proportional to the global spatial RMS) of the difference between the Swarm and GRACE models and the unsmoothed GRACE climatological model, for two levels of smoothing: 750km (Figure 12) and 1500km (Figure 13).

For the 750km case, the Swarm difference has the same amplitude as the climatological model for the period later than 2018. This is largely the result of a positive trend of around 8mm/year in the spatial RMS of the GRACE climatological

model, while the amplitude of the Swarm difference remains roughly constant since 2016. This positive trend is associated

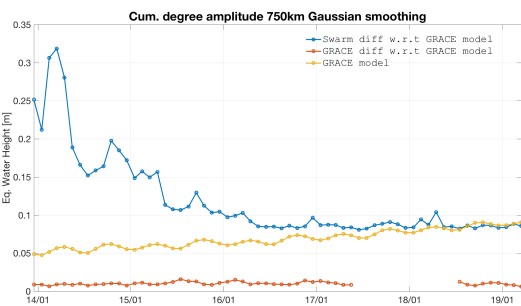

**Figure 12.** Cumulative degree amplitude (or global spatial RMS) of the difference between the Swarm and CSR RL06 GRACE GFMs relative to the GRACE climatological model, as well as the latter, considering 750km Gaussian smoothing.

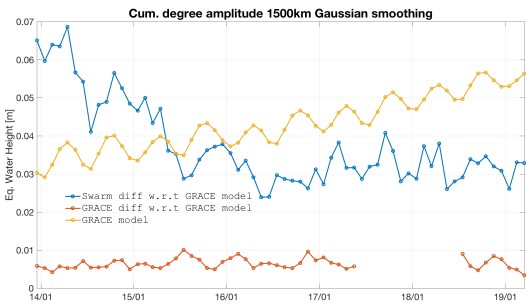

**Figure 13.** Cumulative degree amplitude (or global spatial RMS) of the difference between the Swarm and CSR RL06 GRACE GFMs relative to the GRACE climatological model, as well as the latter, considering 1500km Gaussian smoothing.

with an increased spatial variability represented by the GRACE climatological models, not by long-term trends such as ice-loss in polar regions; we confirmed this interpretation by computing the grid mean and STD separately (not shown). We will demonstrate in Section 3.4.2 that a significant portion of the amplitude of the Swarm difference is located over ocean areas and the agreement over land is significantly better. Therefore, the same amplitude between Swarm and the GRACE climatological

model suggests this smoothing intensity is the minimum required to isolate the geophysical signal in the Swarm time series.

In case of 1500km smoothing, the Swarm differences have lower amplitudes than the GRACE climatological model since mid-2015. We interpret this observation, given the conservative nature of the Swarm global RMS difference, as indication that there is unnecessary suppression of the signal at spatial wavelengths from the two smoothing intensities considered in Figures 12 and 13 (roughly 1500km to 3000km, since we report smoothing radii).

We repeated this exercise also for the cases of no smoothing and 300km smoothing radius. Those results indicated that the errors above degree 12 dominate the solution and produce monthly differences much larger than the amplitude of geophysical signal contained in the GRACE climatological model, at least one order of magnitude for no smoothing and at least a factor of 7 for the 300km case (not shown).





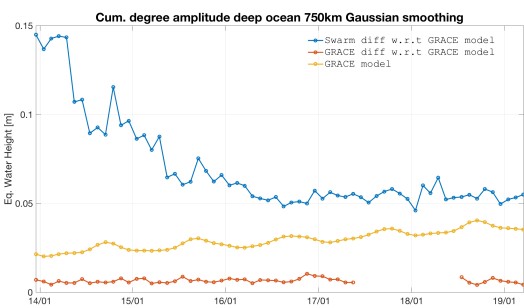

**Figure 14.** Agreement of the combined Swarm monthly solutions and the CSR RL06 GRACE solutions with the GRACE climatological model in terms of cumulative degree amplitude (maximum degree 40) as function of time, for deep ocean areas considering 750km smoothing.

### 3.4.2 Land and deep ocean signal

One important aspect of the Swarm gravity field models is the substantial error in the oceanic regions. We do not have a definitive explanation for this observation, other than the ionospheric activity may corrupt more significantly the estimated gravity field parameters over the oceans since away from land areas there is very little gravity signal to capture. This section illustrates this characteristic, by computing separate statistics for land and deep ocean areas, the latter defined in Section 2.6.2.

In Figure 14 the RMS of the deep ocean areas is shown in terms of the difference between the Swarm and GRACE solutions
relative to the GRACE climatological model, as well as the latter. As expected, the GRACE GFMs differ very little from the climatological model, well under $1$cm EWH. On the other hand, the Swarm GFMs show differences which are of higher amplitude than the ocean signal represented by the climatological model. In other words, the SNR of the Swarm GFMs, as represented by the spatial RMS, is consistently below one over ocean areas.

We illustrate the agreement of Swarm and GRACE solutions with the climatological model in the spectral domain in Fig-
ure 15 (which is produced in a similar way as Figure 10).

Similar to the evolution of the temporal agreement represented in Figure 14, the spectral analysis illustrates that Swarm differs from the climatological model with amplitudes that surpass the signal, across all the spatial wavelengths, over the oceanic areas. The only exception refers to degree 2 but that is mainly driven by the consistent use of $C_{20}$ published in Cheng and Ries (2018).

When it comes to land areas, the Swarm solutions agree with the climatological model much better than in the oceans. Figure 16 shows that since mid-2015, the difference with respect to the climatological model has a smaller amplitude than the signal in the latter. This means that Swarm is generally able to observe the majority mass transport processes described by the climatological model (under Gaussian smoothing with $750$km radius), in particular after 2016. Prior to mid-2015, this is on average not the case although we will demonstrate in Section 3.5.3 that regions where the mass transport signal is of substantial
amplitude are reasonably well observed.

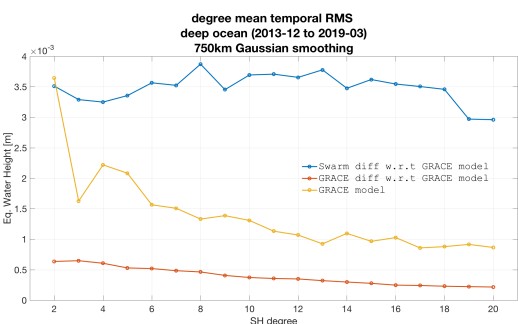

**Figure 15.** Agreement of the combined Swarm monthly solutions and the CSR RL06 GRACE solutions with the GRACE climatological model in terms of the degree mean of the per-coefficient temporal RMS difference, for deep ocean areas considering 750km smoothing.

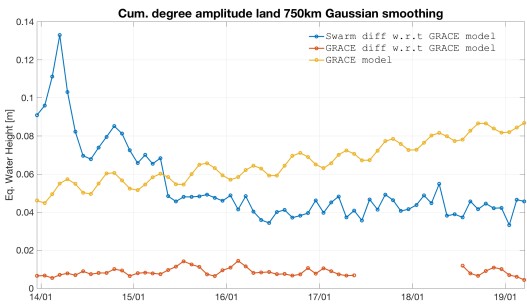

**Figure 16.** Agreement of the combined Swarm monthly solutions and the CSR RL06 GRACE solutions with the GRACE climatological model in terms of cumulative degree amplitude (maximum degree 40) as function of time, for land areas considering 750km smoothing.

The analysis in the spectral domain summarized in Figure 17 confirms that the difference with respect to the climatological model is of smaller amplitude than the signal therein represented up to degree 12. This result further confirms the result of Section 3.4.1 regarding de adequacy of smoothing the Swarm solutions with a Gaussian filter with 750km radius.

We now focus on the necessary smoothing to retrieve any deep ocean signal from the monthly Swarm models. We increased
the smoothing intensity relative to what is discussed in Section 3.4.2 to demonstrate the capabilities of Swarm to contribute to ocean mass studies. We tested smoothing radii of 1000, 1500, 3000 and 5000km; the results for 3000km are presented in Figures 18 and 19.

Figure 18 demonstrates that a smoothing radius of 3000km is enough to reduce the spatial RMS of the difference between Swarm and the GRACE climatological model below the spatial RMS of the latter, particularly after 2016. This means that
since 2016 Swarm has been observing ocean mass changes at the extremely coarse spatial scale of roughly 6000km.

We further demonstrate Swarm's ability to resolve large scale ocean mass changes in the spectral domain, Figure 18. As illustrated, the smoothing radius of 3000km is barely enough to, on average throughout the whole Swarm period, decrease the degree average of the per-degree RMS difference below the signal amplitude, as represented by the adopted climatological



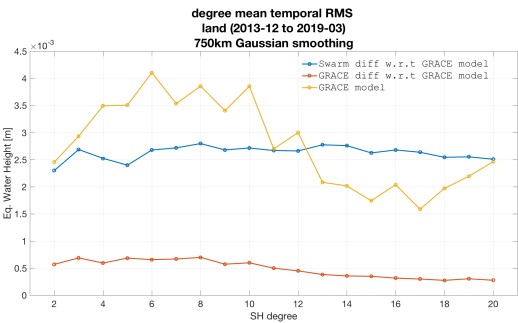

**Figure 17.** Agreement of the combined Swarm monthly solutions and the CSR RL06 GRACE solutions with the GRACE climatological model in terms of the degree mean of the per-coefficient temporal RMS difference, for land areas considering 750km smoothing.

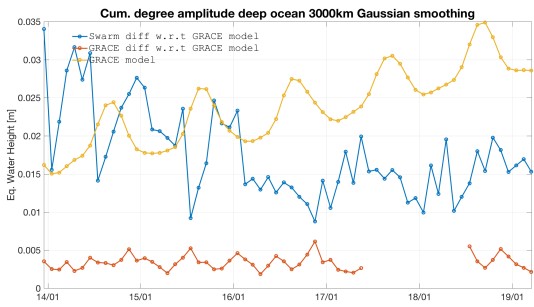

**Figure 18.** Deep ocean spatial RMS of the difference between the Swarm and GRACE models relative to the GRACE climatological model, considering 3000km Gaussian smoothing.

model. Note that the spectral measure represented by the degree average considered the complete Swarm period, including the
start of the mission, when the quality of the solutions was the lowest. Therefore, the smoothing radius of 3000km is well suited
to resolve Swarm deep ocean mass changes since mid-2015.

### 3.5 Signal content

This section describes the geophysical signal represented by the Swarm models. We start by illustrating the time series of
a few low degree coefficients in Section 3.5.1. The variability of the Swarm model, and the patterns therein, is discussed in
Section 3.5.2. We end with Section 3.5.3, where we give an overview of the capabilities of the Swarm models to observe large
basin storage variations and how they compare to GRACE and GRACE-FO.

### 3.5.1 Low degrees

We now present the time series of a selection of low degree coefficients, without any smoothing applied. This section aims at
illustrating in the time domain the noise characteristics of the Swarm models and how they compare to GRACE. We give an



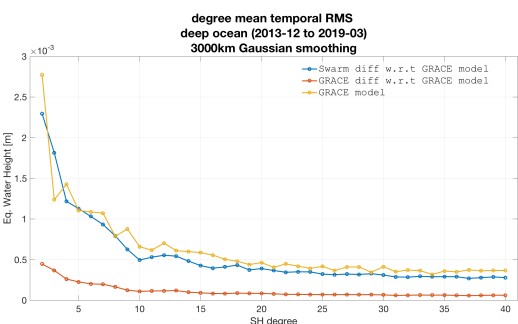

**Figure 19.** Spectral agreement in terms of the degree mean of the per-coefficient temporal RMS difference, of the Swarm solutions and the GRACE solutions with the GRACE climatological model, considering 3000km Gaussian smoothing.

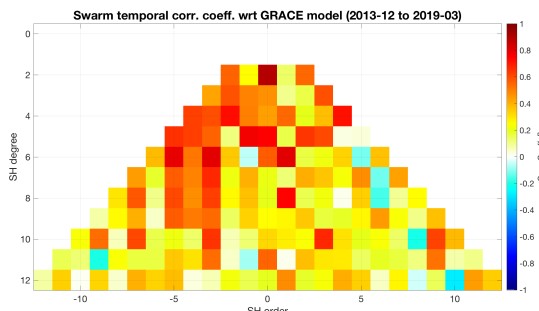

**Figure 20.** Per-coefficient correlation coefficient between the GRACE climatological model and Swarm.

overview of the per-degree correlation coefficients of Swarm and GRACE relative to the climatological model. The degree 2 coefficients (except $C_{2,0}$), which are particular important for Sea-level studies, are subsequently presented. Finally, we show the selected case of $C_{5,-1}$ that has an interesting temporal evolution and how Swarm and GRACE capture those signals. The time series of the zonal coefficients from degrees 3 to 5 are presented in Appendix D. Note that we represent the sine Stokes coefficients with negative order, e.g. $C_{2,-1}$.

Figures 20 and 21 represent the correlation coefficient of the time series of Swarm and GRACE relative to the climatological model, including the early period of the mission when the quality of the Swarm models was lower. As expected, GRACE's coefficients correlated much more closely to the climatological model, as represented by the numerous dark red pixels in the triangular plot of Figure 21. The overview of Swarm's correlation with the climatological model (Figure 20) is dominated by values of around 0.2 (represented by a yellow colour), with some regions with average correlations of roughly 0.6 (represented

by the red colour), notably for orders -5 to -3 and degrees 9 to 4. Furthermore, we observe some interesting common features in both Swarm and GRACE correlation plots, namely order -6 and $C_{5,5}$ seems to be poorly captured by the climatological mode, since neither Swarm nor GRACE correlated well.

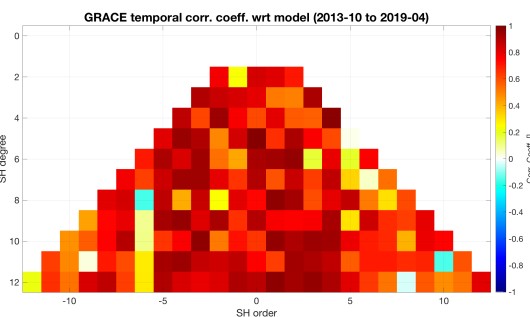

**Figure 21.** Per-coefficient correlation coefficient between the GRACE climatological model and GRACE.

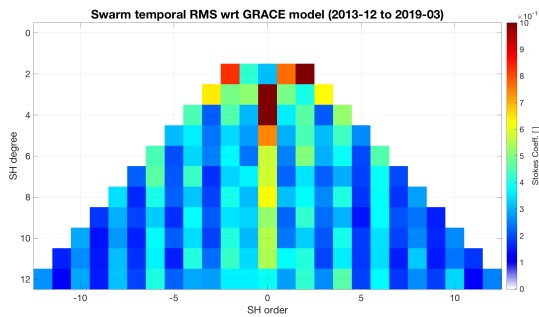

**Figure 22.** Per-coefficient RMS of the difference between the GRACE climatological model and Swarm.

Figure 22 illustrates one particularity of the Swarm models. The RMS of the difference relative to the GRACE climatological model is heavily order dependent, with the even orders showing a larger RMS than the odd orders (for degrees 4 and above); this effect is particularly striking for orders 6 and -6, as well as for 5 and -5. This feature is also present in the individual models (not shown). We cannot find an explanation for the discrepancy between the RMS difference in even and odd orders.

Figures 23 to 26 show the time series of the degree 2 coefficients. They illustrate the general characteristics of Swarm coefficient time series: large signal amplitudes, in particular before mid-2015, as well as a general agreement in the average value, if one could imagine a heavy temporal smoothing operation. The last characteristics, which is extremely common for all the coefficients we have analysed (up to degree 6, not all shown here), find a rare exception in $C_{2,1}$, particularly before 2017. A possible explanation is related to the mean pole model (Wahr et al., 2015), which differs between our Swarm solutions (Appendix C) and CSR RL06 (Bettadpur, 2018). Regarding the agreement of the temporal signal captured by Swarm and that captured by GRACE, it is generally possible to observe that Swarm tends to follow roughly in the same direction, albeit with large month-to-month changes (i.e. larger errors) and with frequent over-shootings before 2016. The large errors are the result of the Swarm solutions exploiting the less accurate kinematic positions as gravimetric observations (in comparison to the much more accurate Inter-Satellite Ranges (ISRs) of GRACE). The errors tend to be larger before 2016, during the period of higher



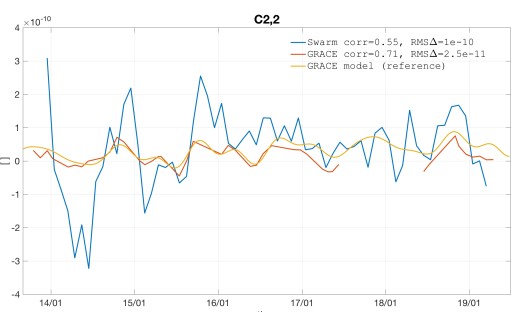

**Figure 23.** Coefficient $C_{2,2}$ as observed by GRACE and Swarm, as well as represented by the GRACE climatological model.

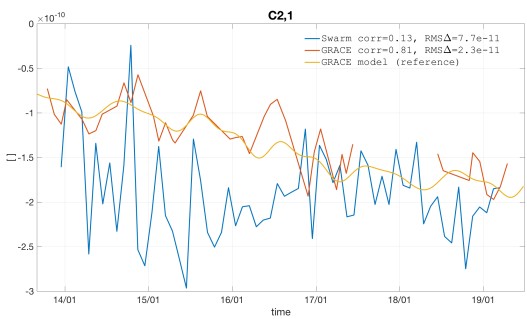

**Figure 24.** Coefficient $C_{2,1}$ as observed by GRACE and Swarm, as well as represented by the GRACE climatological model.

solar/ionospheric activity as well as prior to the GPS receiver tracking loop updates (van den IJssel et al., 2016; Dahle et al., 2017).

Figure 27 shows a representative case of a good agreement between Swarm and GRACE. The overall trend of the $C_{5,-1}$
coefficient is well represented in the climatological model but fails to capture the abnormal deviation around early 2016, which is observed in a consistent way by GRACE and Swarm.

### 3.5.2 Signal variability

The current section is devoted to presenting the signal variability in the Swarm solutions, shown in Figure 28. The most striking feature in the Swarm variability concerns the strong geomagnetic equator signature and the strong artefacts near the
South magnetic pole (due South of Tasmania, on the coast of Antarctica). Interestingly, there is no obvious signature close to the North magnetic pole (North of Hudson bay, West of Greenland). The geomagnetic equator signature extends over land and ocean areas, notably the Saharan desert, although it is possible to distinguish the signature of the strong geophysical signal over the Amazon basin. This artefact is also characterized by an obvious east-west banded structure, which is very well delineated over the central Atlantic, North Africa and Indochina regions. In spite of these artefacts, we will demonstrate that Swarm is
able to resolve monthly large scale mass transport processes. For that purpose we look at the regions circumscribed by the

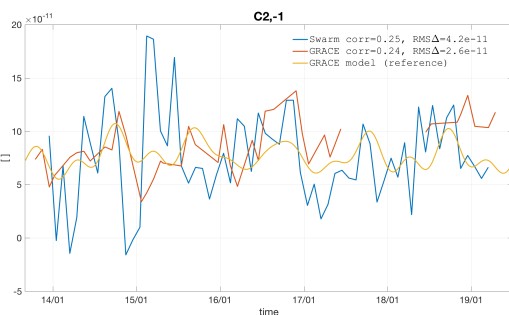

**Figure 25.** Coefficient $C_{2,-1}$ as observed by GRACE and Swarm, as well as represented by the GRACE climatological model.

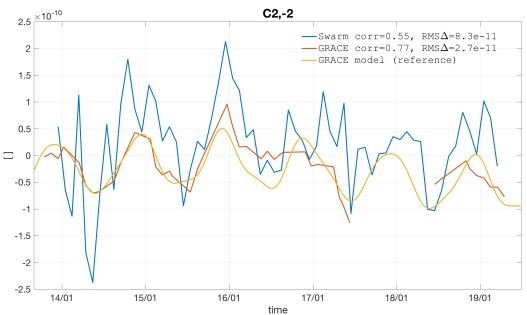

**Figure 26.** Coefficient $C_{2,-2}$ as observed by GRACE and Swarm, as well as represented by the GRACE climatological model.

red dashed rectangles in Figure 28. We choose these regions because they are located at various geographical locations, are of different sizes and are under influence of different types of geophysical signals.

Looking at the variability in the GRACE models over the same periods, Figure 29 (produced in a consistent way as Figure 28), there is no obvious signature of geomagnetic effects. Additionally, the variability over the oceans is very small, in comparison to land areas.

### 3.5.3 Large storage basins

In this section, we present time series of Swarm and GRACE average EWH over the areas highlighted in Figures 28 and 29. Unlike other sections, the GRACE signal (relative to GGM05C) is calculated from the monthly RL06 CSR solutions, after 750km smoothing and the usual $C_{2,0}$ replacement. The trend (and bias) is co-estimated with yearly and semi-yearly sine and cosine periods, in order to be insensitive to phase differences at the beginning and end of the period under analysis. Instead of disclosing the constant term in the polynomial and sinusoidal regression, we prefer to report the average over the period under analysis as measure of a constant bias.

We illustrate these time series with the example of Greenland and Amazon, in Figures 30 and 31, respectively. The remaining time series can be found in Appendix E. As was the case with the analysis of the low degrees, the time series are less smooth

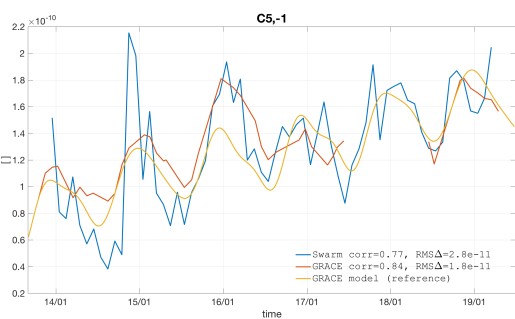

**Figure 27.** Coefficient $C_{5,-1}$ as observed by GRACE and Swarm, as well as represented by the GRACE climatological model.

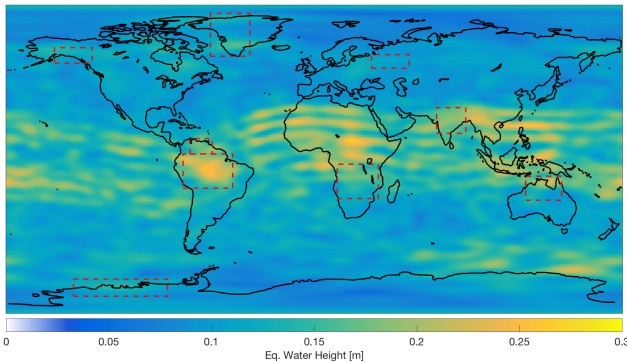

**Figure 28.** Signal variability for Swarm during the period between 2013-12 and 2019-03, under 750km Gaussian smoothing.

than GRACE, as a result of the increased influence of errors. In spite of this, the Swarm time series follows GRACE closely, with a correlation coefficient of 0.79 and 0.95 for Greenland and Amazon, respectively. The trend is over estimated by $-0.3$ and $-1.38 \mathrm{cm/year}$ respectively, mainly as a result of the higher errors before mid-2015. Swarm also agrees with the GRACE-FO observation that the Greenland ice mass loss seems to have slowed down, since both Swarm and GRACE-FO lines are above the linear interpolation. In case of the Amazon basin, the GRACE-FO months agree particularly well with Swarm.

Table 8 provides an overview of the statistics derived from the time series of all analysed basins. The Swarm and GRACE time series agree on their average values between $-1.71 \mathrm{cm}$ (Amazon) and $2.63 \mathrm{cm}$ (Orinoco), on their trend between $-1.38 \mathrm{cm/year}$ (Amazon) and $0.59 \mathrm{cm/year}$ (Congo Zambezi) and on their correlation coefficient between $0.58$ (Congo Zambezi) and $0.95$ (Amazon). All regions show a variety of the values in their statistics, thus making it difficult to immediately identify which one is best observed. For example, although the Amazon time series shows the largest trend difference (in absolute value), it

also has the highest correlation coefficient. The Congo Zambezi basin might be the worst observed location, since it has the largest trend difference (and second in absolute value) and lowest correlation coefficient. In case the period before mid-2015 is ignored, these statistics improve substantially (not shown).



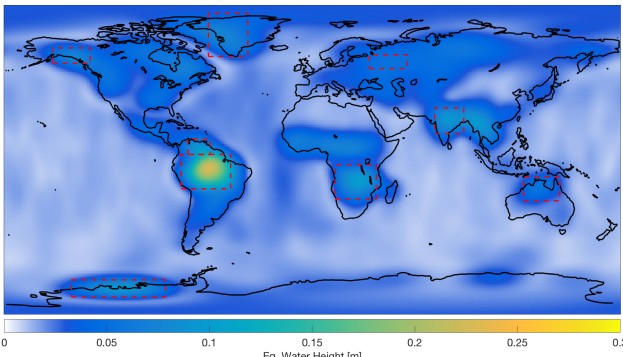

**Figure 29.** Signal variability for GRACE during the Swarm period 2013-12 to 2019-03, including the earliest GRACE-FO solutions, under 750km Gaussian smoothing.

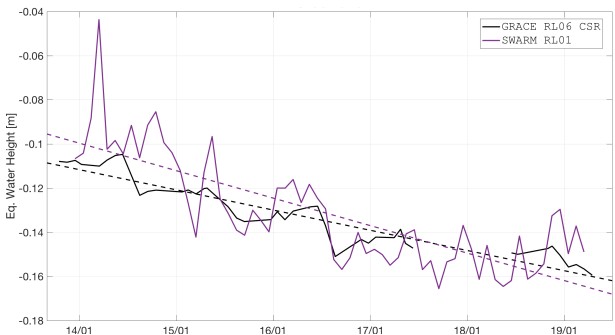

**Figure 30.** Time series of EWH for the Western Greenland region (latitude 60 to 85 degrees, longitude -60 to -37 degrees).

Over the 9 basins presented in this section and in Appendix E, the Swarm RMS difference with respect to GRACE is $1.19\mathrm{cm}$ in terms of temporal mean, $0.60\mathrm{cm/year}$ in terms of trend and shows an average correlation coefficient of 0.75 (bottom of Table 8). Note that the complete Swarm period was considered in deriving these statistics, and represents a conservative estimate of the accuracy of Swarm if the early period before mid-2015 is discarded.





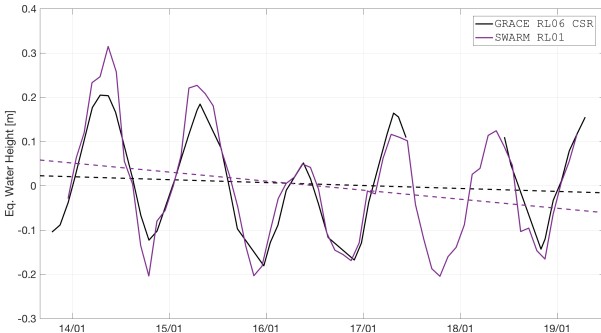

**Figure 31.** Time series of EWH for the Amazon basin (latitude -17 to 3 degrees, longitude -76 to -47 degrees).

**Table 8.** Bias and trend agreement, as well as correlation coefficient between the GRACE and Swarm time series for the selected basins, over the complete Swarm period (December 2013 to March 2019).

| Catchment | temporal mean [cm] | temporal mean $\Delta$ [cm] | linear term [cm/year] | linear term $\Delta$ [cm/year] | corr. coeff. [ ] |
|---|---|---|---|---|---|
| Alaska | -7.73 | -0.23 | -1.84 | -0.38 | 0.76 |
| Amazon | -0.09 | -1.71 | -2.04 | -1.38 | 0.95 |
| Congo Zambezi | 0.84 | -0.70 | 0.25 | 0.59 | 0.58 |
| Ganges-Brahm | -3.27 | 0.35 | -1.42 | -0.04 | 0.72 |
| Greenland | -13.16 | 0.02 | -1.25 | -0.33 | 0.79 |
| N Australia | -0.75 | -1.15 | -1.04 | -0.57 | 0.71 |
| Orinoco | -1.02 | 2.63 | -0.53 | -0.52 | 0.84 |
| Volga | 2.19 | 0.79 | 0.38 | 0.13 | 0.65 |
| W Antarctica | -9.90 | -0.50 | -1.23 | -0.30 | 0.75 |
| Overall | | 1.19 | | 0.60 | 0.75 |

## 4 Conclusions

We present Swarm GFMs resulting from the combination of four individual solutions computed from different gravity field estimation approaches: Celestial Mechanics Approach (CMA), Decorrelated Acceleration Approach (DAA), Improved Energy
Balance Approach (IEBA) and Short-Arcs Approach (SAA). Two approaches (CMA and IEBA) exploit the KO solutions produced at AIUB and the other two (DAA and SAA) the KOs produced at IfG. The combination is done at the solution level, weighted by VCE; for the sake of brevity, we refer to Teixeira da Encarnação and Visser (2019) to demonstrate that our combination produces Swarm models in better agreement with GRACE than if the combination is done at the NEQ level.

We test the added value of KB in the quality of the Swarm GFM, when compared to the long wavelength signal recovered
by GRACE, by computing 7 GFMs during periods of different data quality. We demonstrate that the largest changes in the





results appear during early 2015 (high ionospheric activity, before improvements in Swarm's GPS receivers) that translate into a slight deterioration of the quality of the Swarm solutions, cf. Table 6. For the 5 months analysed in 2016, considering two KO solutions, any improvement is either minimal (0.1 to 0.2mm geoid height, in 5 cases), negligible (in 2 cases) or slightly worse (0.1mm in 3 cases). We conclude that any common errors that would be eliminated in the KB solutions are already (mostly)

corrected in the KOs. For this reason, our Swarm GFMs do not consider KBs.

Another test regarding the added value of additional data took the form of including Swarm-C non-gravitational accelerations. We compared the three-satellite Swarm solution produced considering the DAA and non-gravitational accelerations acting on Swarm-C represented by the TUD and ASU non-gravitational acceleration models, in addition to exploiting the accelerometer measurements. Since the Swarm A and B satellites do not produce usable accelerometer readings, they are rep-

resented by the ASU model exclusively. The results indicate that the accelerometer observations are only beneficial in those cases when the amplitude of the non-gravitational accelerations acting on Swarm-C are of higher amplitude than in quiet periods in solar activity, such as is the case since 2016. This may be the result of the potentially lower quality of the calibrated accelerations, caused by the lower SNR in the accelerometer observations.

Regarding the topic of non-gravitational accelerations in the processing of GPS-driven GFMs, we would like to comment

on the results of Ditmar et al. (2006) and Ditmar et al. (2008), who demonstrated that non-gravitational accelerations are not needed for gravity field estimation and the quality of the GPS observations (and the resulting KOs) are the main drivers of the quality of the GFMs. Within out project, each AC is free to elect whatever processing strategy they deem to be most beneficial to their individual solutions, which is assessed internally. For example, AIUB has determined that the use of daily and 15 minutes piecewise-constant empirical parametrization does not require any modelling of non-gravitational accelerations.

In case of ASU, who exploits a dedicated decorrelation procedure (which is a frequency-dependent noise *whitening* procedure), their solutions benefit from drag, EARP and EIRP models. Essentially, the inclusion of Frequency-Dependent Data Weighting (FDDW) is not within immediate reach to all ACs, in which case other processing strategies seem to produce comparable solution quality. In summary, we do not wish to contest previous results on this topic, but clarify the differences in our processing choices.

We quantify the different quality of the various individual solutions and demonstrate that all have the potential to contribute positively to the quality of the combined Swarm time series of GFMs. We additionally explain that our approach to combine the individual GFMs at the solution level considering VCE weights is an effective way of overcoming the difficulty in combining solutions at the NEQ level when the corresponding normal matrices represent errors of different type, formal and calibrated in our case (Teixeira da Encarnação and Visser, 2019).

For the combined models, we demonstrate that a Gaussian smoothing of 750km radius is necessary to ensure a SNR larger than 1. In case of a more intense smoothing, we tested the case of 1500km, the temporal evolution of the spatial RMS showed a SNR larger than one for the period after mid-2015. As a result of the known errors of Swarm over the ocean, this large SNR indicates there is unnecessary smoothing of the Swarm signal. Masking the Swarm data separately over ocean and land, we demonstrate that Swarm's ability to measure land mass transport processes, with a SNR ratio higher than one for the post

mid-2015 period and 750km smoothing radius. Over the ocean areas, the spatial RMS of the difference between Swarm and



the GRACE climatological model is always larger than the spatial RMS of the latter. To resolve the oceanic signal, the Swarm data required a more intense Gaussian smoothing, with a radius of 3000km.

We analyse the signal content of the Swarm models in terms of time series of the low degrees, spatial patterns of the temporal signal variability and time series of large storage basins. Comparing the time series of isolated SH coefficients,
we show that the Swarm data generally has a more erratic temporal evolution with sudden month-to-month variations. We attribute this particularity to the lower accuracy of the GPS observations as gravimetric data, as compared to GRACE's K-Band Ranging (KBR) data. We also illustrate features in the Swarm data that are not captured by the climatological model, but confirmed by the GRACE/GRACE-FO data, notably the atypical deviation around early 2016 in the $C_{5,-1}$ coefficient and an apparent phase shift in the $C_{3,0}$ coefficient after 2017. By plotting the spatial patterns in the temporal variability of the
Swarm data, we bring into evidence the strong signature over the geo-magnetic equator, showing strong meridional stripes, and over the South Magnetic Pole (but not on the North Magnetic Pole). In spite of this artefact, the strong mass variability over the Amazon basin is clearly visible. In what regards the time series of mass changes over large storage basins, Swarm agrees on average with GRACE (the climatological model was not relevant to this analysis) at $1.19$cm in terms of temporal mean, $0.60$cm/year in terms of trend and 0.75 correlation coefficient over the 9 basins we considered. We show that Swarm agrees
with the observation of GRACE-FO that the ice mass loss over Greenland seems to have slowed down during late 2018, in spite of the heavy signal dilution caused by the necessary smoothing to reduce the errors in the Swarm models.

Although our Swarm models are already in a production mode, we are considering several options to improve their quality. Given the high sensitivity of the KOs to ionospheric activity, we plan to focus our efforts to improve the weighting of the GPS observations (Dahle et al., 2017; Kermarrec et al., 2018; Schreiter et al., 2019). We also plan to decrease the disagreement
between the individual solution produced at OSU and those at other ACs by including advanced algorithms for reducing the effects of jumps and the amplification of high-frequency noise in the differentiation of the KO positions into velocities.

*Data availability.* The Swarm monthly models are distributed on a quarterly basis at ESA's Earth Online Swarm Data Access (at https://swarm-diss.eo.esa.int/#swarm, follow *Level2longterm* and then *EGF*) and at the International Centre for Global Earth Models (http://icgem.gfz-potsdam.de/series/03_COST-G/Swarm), as well as identified with the DOI 10.5880/ICGEM.2019.006 (Encarnacao et al., 2019).

**Appendix A: Kinematic Orbits**

**A1  Delft University of Technology**

| | |
|---|---|
| **Software:** | GPS High precision Orbit determination Software Tool (van Helleputte, 2004; Wermuth et al., 2010) |
| **Differencing Scheme:** | Undifferenced |
| 700   **Linear combination:** | Ionosphere-free |
| **GPS observations:** | Code and carrier phase |



| | |
|---|---|
| **Estimator:** | Bayesian weighted LS |
| **Arc length:** | 30 hours |
| **Data weighting:** | a-priori weights equal to 1m and 1mm for code and phase observations (resp.) |
| **Transmitter PCV:** | Official IGS08 ANTEX up to day 17/028, official IGS14 ANTEX from day 17/029 on |
| **Receiver PCV:** | empirically determined from stacking of reduced-dynamic POD residuals with 1° binning |
| **Data screening:** | minimum SNR of 10, minimum of 6 GPS satellites, code and phase outlier editing threshold of 2m and 3.5cm, respectively, 1 meter or larger difference between estimated KO positions and with Reduced-Dynamic Precise Science Orbit (PSO) |
| **Earth precession model:** | International Astronomical Union (IAU) 1976 (Lieske et al., 1977) |
| **Earth nutation model:** | IAU 1980 (Seidelmann, 1982) |
| **Earth orientation model:** | Centre for Orbit Determination in Europe (CODE) final Earth Rotation Parameters (ERP) |

### A2 Astronomical Institute of the University of Bern

| | |
|---|---|
| **Software:** | Bernese v5.3 (Dach et al., 2015; Jäggi et al., 2006) |
| **Differencing Scheme:** | Undifferenced |
| **Linear combination:** | Ionosphere-free |
| **GPS observations:** | Carrier phase |
| **Estimator:** | Batch LS |
| **Arc length:** | 24 hours |
| **Data weighting:** | Not Applicable (N/A) |
| **Transmitter PCV:** | Official IGS08 ANTEX up to day 17/028, official IGS14 ANTEX from day 17/029 on |
| **Receiver PCV:** | Stacking of residuals from reduced-dynamic Precise Orbit Determination (POD) of approx. 120 days, 9 iterations, 1° binning |
| **Data screening:** | 2cm/s or larger time-differences of the geometry-free linear combination of L1B GPS carrier phase observations |
| **Earth precession model:** | International Earth Rotation Service (IERS) 2010 Conventions (Petit and Luzum, 2010) |
| **Earth nutation model:** | IERS 2010 Conventions (Petit and Luzum, 2010) |
| **Earth orientation model:** | CODE final ERP |

### A3 Institute of Geodesy Graz

| | |
|---|---|
| **Software:** | GROOPS |
| **Differencing Scheme:** | None |
| **Linear combination:** | None (the ionospheric influence is co-estimated) |
| **GPS observations:** | Code and carrier phase |
| **Estimator:** | LS |



| | |
|---|---|
| **Arc length:** | 24 hours |
| **Data weighting:** | Elevation and azimuth-dependent, epoch-wise VCE |
| **Transmitter PCV:** | Empirical, estimated from 5.5 years of data, including data from several LEO missions (GRACE, Jason 2 & 3, MetOp-A & -B, Sentinel 3A, Swarm, TanDEM-X, TerraSAR-X) (Zehentner, 2016) |
| **Receiver PCV:** | Empirical, spherical harmonics (maximum D/O 60), derived from 38 months of data |
| **Data screening:** | Implicit in VCE |
| **Earth precession model:** | IAU 2006/2000A precession-nutation model (Petit and Luzum, 2010) |
| **Earth nutation model:** | IAU 2006/2000A precession-nutation model (Petit and Luzum, 2010) |
| **Earth orientation model:** | IERS Earth Orientation Parameter (EOP) 08 C04 (Petit and Luzum, 2010) |

### A4 Common

| | |
|---|---|
| **Carrier phase ambiguities:** | Float |
| **Receiver clock corrections:** | Co-estimated |
| **Sampling rate:** | 10 or 1 seconds (depending on L1B GPS data) |
| **Elevation cut-off angle:** | 0° |
| **GPS orbits and clocks:** | Final orbits and 5 seconds clocks of Centre for Orbit Determination in Europe (Dach et al., 2017) |
| **Swarm attitude:** | L1B attitude data |

### Appendix B: Kinematic Baselines

### B1 Delft University of Technology

| | |
|---|---|
| **Software:** | Multiple satellites Orbit Determination using Kalman filtering (van Barneveld, 2012) |
| **Linear combination:** | N/A (the ionospheric frequency-dependent influence is modelled) |
| **Estimator:** | Iterative EKF |
| **Carrier phase ambiguities:** | Integer, using the Least-squares Ambiguity De-correlation Adjustment method (Teunissen, 1995) |
| **Receiver PCV:** | Empirical Phase Center Variations (PCVs) and Code Residual Variations (CRVs) maps are estimated a priori for each GPS frequency |

### B2 Astronomical Institute of the University of Bern

| | |
|---|---|
| **Software:** | Bernese (Dach et al., 2015; Jäggi et al., 2006), development version |
| **Linear combination:** | Ionosphere-free |




| 765 | **Estimator:** | LS |
| | **Carrier phase ambiguities:** | wide-lane and narrow-lane integer ambiguity fixing with the Melbourne-Wübbena and the ionosphere-free linear combination, respectively |
| | **Receiver PCV:** | Empirical |

### B3   Common

| 770 | **Differencing Scheme:** | Double-differenced |
| | **GPS observations:** | Code and carrier phase |
| | **Carrier phase ambiguities:** | Integer |

### Appendix C:  Gravity Field Models

### C1   Astronomical Institute of the University of Bern

| 775 | **Software:** | Bernese v5.3 (Dach et al., 2015; Jäggi et al., 2006) |
| | **Approach:** | Celestial Mechanics Approach (Beutler et al., 2010) |
| | **Reference GFM:** | AIUB GRACE-only static model, version 3 (Jäggi et al., 2011) |
| | **Empirical Parameters:** | Daily and 15 minutes piecewise-constant (constrained) |
| | **Drag Model:** | None |
| 780 | **EARP and EIRP Models:** | None |
| | **Non-tidal Model:** | Atmosphere and Ocean De-aliasing Level 1B (Flechtner, 2011) |
| | **Ocean Tidal Model:** | 2011 Empirical Ocean Tide model (Savcenko and Bosch, 2012) |
| | **Permanent Tide System:** | tide-free |

### C2   Astronomical Institute Ondřejov

| 785 | **Software:** | (developed in-house) |
| | **Approach:** | Decorrelated Acceleration Approach (Bezděk et al., 2014) |
| | **Reference GFM:** | ITG GRACE-only static model, 2010 (Mayer-Gürr et al., 2010) |
| | **Empirical Parameters:** | Daily constant-piecewise |
| | **Drag Model:** | Naval Research Laboratory Mass Spectrometer and Incoherent Scatter Radar (Picone et al., 2002) |
| 790 | | |
| | **EARP and EIRP Models:** | Knocke et al. (1988) |
| | **Non-tidal Model:** | Atmosphere and Ocean De-aliasing Level 1B (Dobslaw et al., 2017) |
| | **Ocean Tidal Model:** | 2004 Finite Element Solution (Lyard et al., 2006) |
| | **Permanent Tide System:** | tide-free |





## C3    Institute of Geodesy Graz

| | |
|---|---|
| **Software:** | GROOPS |
| **Approach:** | Short-Arcs Approach (Mayer-Gürr, 2006) |
| **Reference GFM:** | GOCO release 05 satellite-only gravity field model (Mayer-Gürr, 2015) |
| **Empirical Parameters:** | Piecewise linear for each arc (ranging from 15 to 45 minutes) |
| **Drag Model:** | Jacchia-Bowman 2008 (Bowman et al., 2008) |
| **EARP and EIRP Models:** | Rodriguez-Solano et al. (2012) |
| **Non-tidal Model:** | Atmosphere and Ocean De-aliasing Level 1B RL06 (Dobslaw et al., 2017) |
| **Ocean Tidal Model:** | 2014 Finite Element Solution (Carrere et al., 2015) |
| **Permanent Tide System:** | zero tide |

## C4    Ohio State University

| | |
|---|---|
| **Software:** | (developed in-house) |
| **Approach:** | Improved Energy Balance Approach (Shang et al., 2015) |
| **Reference GFM:** | GRACE Intermediate Field 48 (Ries et al., 2011) up to Degree and Order (D/O) 200 |
| **Empirical Parameters:** | 2nd order polynomial every 3 hours, 1-CPR sinusoidal every 24 hours |
| **Regularization:** | none |
| **Drag Model:** | Naval Research Laboratory Mass Spectrometer and Incoherent Scatter Radar (Picone et al., 2002) |
| **EARP and EIRP Models:** | Knocke et al. (1988) |
| **Non-tidal Model:** | Atmosphere and Ocean De-aliasing Level 1B (Flechtner, 2011) |
| **Ocean Tidal Model:** | 2011 Empirical Ocean Tide model (Savcenko and Bosch, 2012) |
| **Permanent Tide System:** | tide-free |

## C5    Common

| | |
|---|---|
| **Atmospheric Tidal Model:** | Biancale and Bode (2006) |
| **Solid Earth Tidal Model:** | IERS2010 |
| **Pole Tidal Model:** | IERS2010 |
| **Ocean Pole Tidal Model:** | IERS2010 |
| **Third body perturbations:** | Sun, Moon, Mercury, Venus, Mars, Jupiter and Saturn, following the JPL Planetary and Lunar Ephemerides (Folkner et al., 2014) |
| $C_{2,0}$ **coefficient:** | estimated alongside other coefficients |


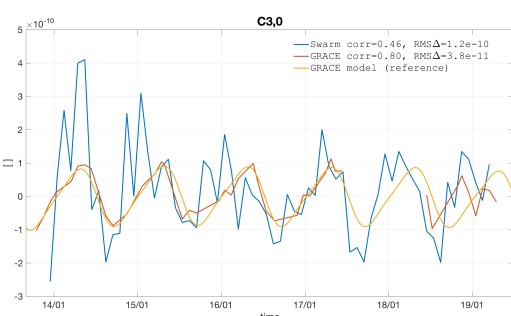

**Figure D1.** Coefficient $C_{3,0}$ as observed by GRACE and Swarm, as well as represented by the GRACE climatological model.

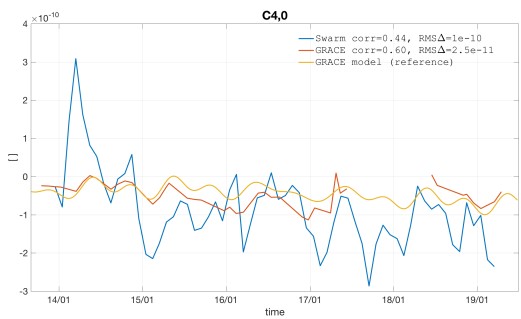

**Figure D2.** Coefficient $C_{4,0}$ as observed by GRACE and Swarm, as well as represented by the GRACE climatological model.

825 **Appendix D: Time series of zonal coefficients**

Figures D1 to D3 illustrate the time series for the zonal coefficients of degrees 3 to 6, respectively.

The zonal coefficient of degree 3 is an interesting case because both Swarm and GRACE-FO observe a phase shift during late 2018, relative to the climatological model, which is well in-phase with GRACE for the non GRACE-FO period (2003 to 2017). Swarm already captures this phase shift possibly as early as mid-2017, although the noisy character of the Swarm time
830 series weakens this type of statement.

The zonal coefficient of degree 4 is one of the few examples where the Swarm time series shows a clear bias relative to GRACE and the climatological model, after 2017 in this case. As was the case for $C_{2,1}$, we cannot explain such behaviour.

The zonal coefficient of degree 5 is an example of excellent agreement between all three time series. Swarm still shows the characteristic noise, as well as a higher overall disagreement before mid-2015. These are features intrinsic to our Swarm
835 solutions.





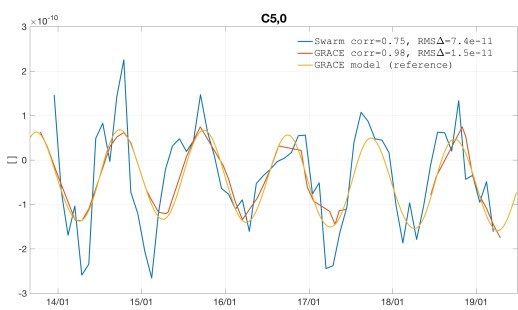

**Figure D3.** Coefficient $C_{5,0}$ as observed by GRACE and Swarm, as well as represented by the GRACE climatological model.

**Appendix E: Storage basin time series**



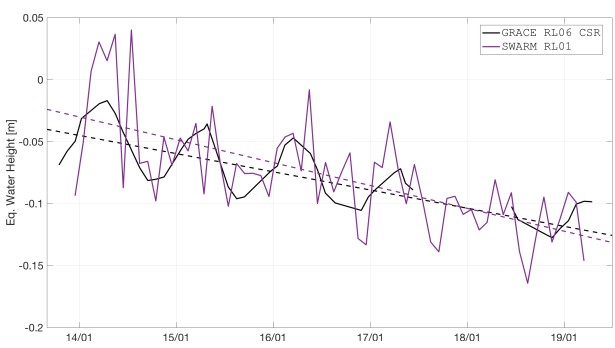

**Figure E1.** Time series of EWH for the Alaska (latitude 56 to 65 degrees, longitude -151 to -129 degrees).

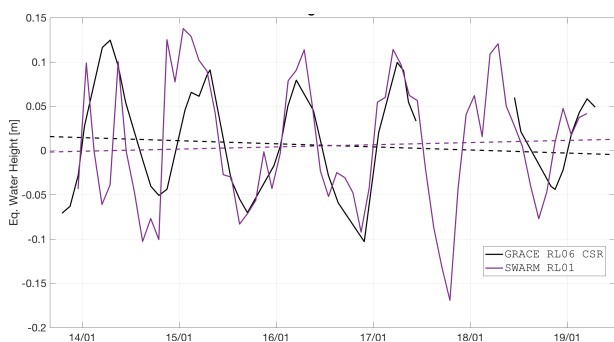

**Figure E2.** Time series of EWH for the Congo and Zambezi basins (latitude -23 to -3 degrees, longitude 14 to 38 degrees).

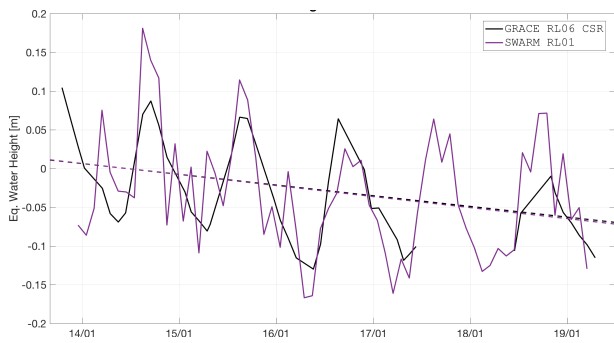

**Figure E3.** Time series of EWH for the Ganges-Brahmaputra basin (latitude 15 to 30 degrees, longitude 72 to 89 degrees).



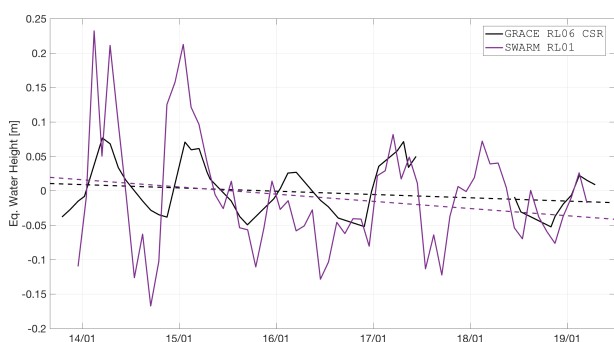

**Figure E4.** Time series of EWH for the Northern Australia region (latitude -24 to -10 degrees, longitude 124 to 145 degrees).

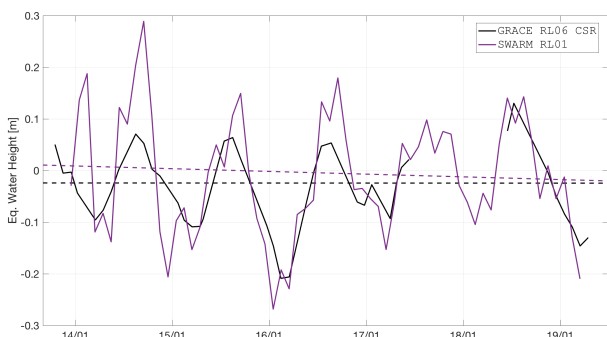

**Figure E5.** Time series of EWH for the Orinoco basin (latitude -3 to 12 degrees, longitude -72 to -59 degrees).

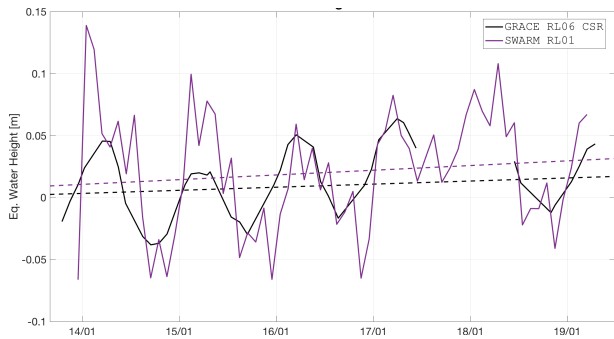

**Figure E6.** Time series of EWH for the Volga basin (latitude 53 to 61 degrees, longitude 34 to 56 degrees).





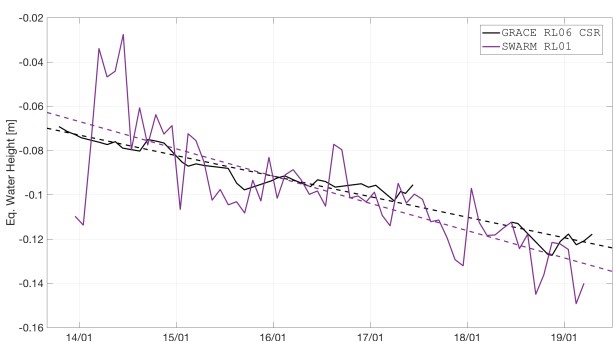

**Figure E7.** Time series of EWH for the Western Antarctica region (latitude -80 to -70 degrees, longitude -140 to -85 degrees).



*Competing interests.* No competing interests are present.

*Acknowledgements.* This research was funded by the European Space Agency (contracts SW-CO-DTU-GS-111 and SW-CN-DTU-GS-027,
part of contract 4000109587/13/I-NB), partially supported by the Strategic Priority Research Program of the Chinese Academy of Sciences
840    (grant XDA19070302), by the National Natural Science Foundation of China (Grant 41584016) and by the Ministry of Education, Youth
and Sports of the Czech Republic (Grant LTT18011). We thank the Swarm Data, Innovation and Science Cluster (DISC) for the support,
flexibility and guidance during the project activities. We also thank Dr. Mark Tamisiea for the fruitful discussions and critical viewpoints.



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

## Acronyms

| | |
|---|---|
| **3D** | three-dimensional |
| **AA** | Acceleration Approach |
| **DAA** | Decorrelated Acceleration Approach |
| **AC** | Analysis Center |
| **AIUB** | Astronomical Institute of the University of Bern, Switzerland |
| **AIUB-GRACE03S** | AIUB GRACE-only static model, version 3 |
| **AOD1B** | Atmosphere and Ocean De-aliasing Level 1B |
| **AOD1B-RL06** | Atmosphere and Ocean De-aliasing Level 1B RL06 |
| **ANGARA** | Analysis of Non-Gravitational Accelerations due to Radiation pressure and Aerodynamics |
| **ASU** | Astronomical Institute (Astronomický ústav), AVCR, Ondřejov |
| **AVCR** | Czech Academy of Sciences (Akademie věd České Republiky), Czech Republic |
| **CHAMP** | CHallenging Mini-Satellite Payload |
| **CODE** | Centre for Orbit Determination in Europe |





| | | |
|---|---|---|
| | **CMA** | Celestial Mechanics Approach |
| | **CoM** | Centre of Mass |
| 1105 | **COST-G** | Combination Service of Time-variable Gravity Fields |
| | **CPR** | Cycle Per Revolution |
| | **CRV** | Code Residual Variation |
| | **CSR** | Center for Space Research, UTexas, USA |
| | **D/O** | Degree and Order |
| 1110 | **DAA** | Decorrelated Acceleration Approach |
| | **DD** | Double-differenced |
| | **DISC** | Data, Innovation and Science Cluster |
| | **DOI** | Digital Object Identifier |
| | **DWM07** | Disturbance Wind Model 07 |
| 1115 | **EARP** | Earth Albedo Radiation Pressure |
| | **EGSIEM** | European Gravity Service for Improved Emergency Management, EU Horizon 2020 |
| | **EIRP** | Earth Infrared Radiation Pressure |
| | **EKF** | Extended Kalman Filter |
| | **EBA** | Energy Balance Approach |
| 1120 | **EOT** | Empirical Ocean Tide model |
| | **EOT11a** | 2011 Empirical Ocean Tide model |
| | **EWH** | Equivalent Water Height |
| | **EOP** | Earth Orientation Parameter |
| | **ERBE** | Earth Radiation Budget Experiment |
| 1125 | **ERP** | Earth Rotation Parameters |
| | **ESA** | European Space Agency |
| | **EU** | European Union |
| | **FDDW** | Frequency-Dependent Data Weighting |
| | **FES** | Finite Element Solution global tide model |
| 1130 | **FES2004** | 2004 Finite Element Solution |
| | **FES2014** | 2014 Finite Element Solution |
| | **GFM** | Gravity Field Model |
| | **GHOST** | GPS High precision Orbit determination Software Tool |
| | **GGM05C** | Combined GRACE Gravity Model 05 |
| 1135 | **GIF48** | GRACE Intermediate Field 48 |
| | **GOCE** | Gravity field and steady-state Ocean Circulation Explorer |
| | **GOCO** | Gravity Observation COmbination |
| | **GOCO05S** | GOCO release 05 satellite-only gravity field model |




| | | |
|---|---|---|
| | **GPS** | Global Positioning System |
| 1140 | **GRACE** | Gravity Recovery And Climate Experiment |
| | **GRACE-FO** | GRACE Follow On |
| | **GROOPS** | Gravity Recovery Object Oriented Programming System |
| | **hl-SST** | High-low Satellite-to-Satellite tracking |
| | **HWM07** | Horizontal Wind Model 07 |
| 1145 | **IAG** | International Association of Geodesy |
| | **IAU** | International Astronomical Union |
| | **ICGEM** | International Centre for Global Earth Models |
| | **IEBA** | Improved Energy Balance Approach |
| | **IERS** | International Earth Rotation Service |
| 1150 | **IERS2010** | IERS Conventions 2010 |
| | **IfG** | Institute of Geodesy, TUG, Graz |
| | **IGFS** | International Gravity Field Service |
| | **IR** | Infrared Radiation |
| | **ISR** | Inter-Satellite Range |
| 1155 | **ITG** | Institut für Geodäsie und Geoinformation, Germany |
| | **ITSG** | Institute of Theoretical Geodesy and Satellite Geodesy |
| | **ITG-GRACE2010s** | ITG GRACE-only static model, 2010 |
| | **ITSG-GRACE2016** | ITSG GRACE-only model, 2016 |
| | **JB2008** | Jacchia-Bowman 2008 |
| 1160 | **JPL** | Jet Propulsion Laboratory, USA |
| | **JPL-PLE** | JPL Planetary and Lunar Ephemerides |
| | **KB** | Kinematic Baseline |
| | **KBR** | K-Band Ranging |
| | **KO** | Kinematic Orbit |
| 1165 | **L1A** | Level 1A data |
| | **L1B** | Level 1B data |
| | **L2** | Level 2 data |
| | **L2PS** | Level 2 Processing System |
| | **LAMBDA** | Least-squares Ambiguity De-correlation Adjustment |
| 1170 | **LEO** | Low-Earth Orbit |
| | **ll-SST** | low-low Satellite-to-Satellite Tracking |
| | **LoS** | Line of Sight |
| | **LS** | Least-Squares |
| | **MAD** | Median Absolute Deviation |





| 1175 | **MODK** | Multiple satellites Orbit Determination using Kalman filtering |
| | **N/A** | Not Applicable |
| | **NEQ** | Normal Equation |
| | **NRLMSISE** | US Naval Research Laboratory Mass Spectrometer and Incoherent Scatter Radar Atmospheric model |
| | **NRTDM** | Near Real-Time Density Model |
| 1180 | **OSU** | Ohio State University |
| | **PCV** | Phase Center Variation |
| | **POD** | Precise Orbit Determination |
| | **PSO** | Precise or Post-processed Science Orbit |
| | **RINEX** | Receiver Independent Exchange Format |
| 1185 | **RL05** | Release 5 |
| | **RL06** | Release 6 |
| | **RMS** | Root Mean Squared |
| | **SAA** | Short-Arcs Approach |
| | **SC** | Stokes coefficient |
| 1190 | **SH** | Spherical Harmonic |
| | **SLR** | Satellite Laser Ranging |
| | **SNR** | Signal-to-Noise Ratio |
| | **SRP** | Solar Radiation Pressure |
| | **STD** | STandard Deviation |
| 1195 | **TUD** | Delft University of Technology, Netherlands |
| | **TUG** | Graz University of Technology, Austria |
| | **UTexas** | University of Texas at Austin |
| | **USA** | United States of America |
| | **VEA** | Variational Equations Approach |
| 1200 | **VCE** | Variance Component Estimation |
| | **ZD** | Zero-differenced |