# Peer review of "Multi-approach gravity field models from Swarm GPS data"

_Earth System Science Data, 2019_

## Referee Comment (RC1) · Anonymous Referee #1 · 8 Nov 2019

Summary: The paper assesses the utility of gravity fields derived from Swarm to bridge the gap between GRACE and GRACE-FO, as well as potentially fill in missing months from the GRACE and GRACE-FO timeseries. The authors describe the approach to combine four independent gravity field solutions using VCE methods, and assess the utility of including accelerometer data (rather than relying on models of non-gravitational accelerations), as well as kinematic baselines in their processing.

General Review: The paper is well written and comprehensive. It covers important topics, in particular the use of the accelerometer data as well as kinematic baselines in the data processing. I recommend publication with only very minor revisions. One general comment is that the figures are not of high quality – the legends and axes are all very difficult to read. Further, there are many places where multiple figures could be

combined into a single figure with subpanels. This would likely increase readability.

Minor comments:

Section 2.6.1 – It is unclear to me why the authors decided to use this approach for C20. They state that the C20 coefficient available from CSR is only available at the GRACE epochs. However, the same group at CSR produces a monthly 5x5 gravity field solution from SLR, which includes as estimate of C20. These estimates are produced in even calendar month intervals. Why not use this?

Line 323: Typo: "if" should be "to"

Line 328: It is unclear what the authors mean by "These periods drive the orbital inclination of the GRACE satellites...". Perhaps you mean the other way around, i.e., that the orbital inclination drives the tidal aliasing periods? This would be more appropriate.

Line 389: Typo – "which is less straightforward"

Line 511: The authors state it is a mystery as to why the oceans have larger errors than over land. Is this really true? A comparison between Figure 14 and Figure 16 reveals that the error over the ocean is only about ∼25% higher than that over land. It is true that the signal to noise ratio is much lower over the ocean – this is why larger smoothing radii are required – but that seems mostly due to the lower signal amplitudes rather than the higher noise values. Have you tried assessing errors as a function of latitude rather than over different geophysical domains? I wonder if errors actually scale with latitude (Figure 28 would support this), and errors are slightly larger over the ocean simply because of sampling bias as a function of latitude?

Line 652: typo – "out" should be "our"

---

## Author Comment (AC1) · 20 Nov 2019

Unlike what is stated in the list of authors of the current version of this article, please be advised that Christoph Dahle is affiliated with the Astronomical Institute of the University of Bern, in addition to the GFZ German Research Centre for Geosciences.

João Encarnação

---

## Referee Comment (RC2) · Anonymous Referee #2 · 21 Nov 2019

The authors combine several instrument data sets and analysis methods to provide a series of gravity fields for the Swarm mission. Results of a multi-analysis combination solution are compared to individual ones and to GRACE results. The paper provides several methodological analyses, new approaches explored, and it describes intermediate data sets such as kinematic orbits and the accelerometry/non-conservative force modelling. The authors, apparently, also have set up standardized and officialized workflows which is encouraging.

The paper is well-written, very detailed and in-depth on a technical level. As they combine four individual solutions, naturally several analyses have been published already by the individual groups, so there is some overlap. The paper could be probably shortened with respect to the description of the background. But these are minor issues.

[Figure]

As a methodological paper, the paper somewhat lacks a hypothesis. It is clear that by combining many analysis we will have a smoother result. What is expected from the outset? That could have been described better. Which is a pity since they seem to have added interesting new methods. What message is conveyed in view of other LEO missions that could be used for gravity retrievals? Should one have different orbits, instruments, what did we learn now for the next LEO mission?

The big issue for this reviewer is whether the authors were well-advised to submit to ESSD. ESSD focuses on "original research data (sets), furthering the reuse of high-quality data of benefit to Earth system sciences". Here, the focus is clearly on the methodology of generating the data and neither its use nor reuse, and I guess other journals are more appropriate. While the authors motivate their study with the need of the community to rely on data sets for studying "glacial cycles and long-term trends", the GRACE-GRACE-FO gap is 10 months and this is the period where these data will be relevant, in addition to few monthly gaps. It seems like a huge effort and the groups are to be congratulated, but they don't show what Earth Science applications will be enabled now that were not possible. We don't learn from their results for the understanding of processes. In their words, the "consequences of the 10-month gap" are not outlined and it is not clear what we gain.

The other very major problem for this reviewer is that apparently no independent validation is provided, except comparisons to GRACE. We don't have GRACE them for the gap, but the authors could compare their low-degree results to satellite laser ranging (SLR) solutions, e.g. for single coefficients or for the ocean mass change. The authors mention only one SLR time series for C20 but that's just one data set and more exist. Others provided C02 timeseries and it is not clear to this reviewer why they rely on GRACE extrapolatons for this. Moreover, the authors, e.g. in Berne or Potsdam, could easily use their Swarm models in SLR range residual analysis as a validation but it is not done. And several other validation techniques and data sets were developed for GRACE but nothing is used here. This is somewhat disappointing.

---

## Editor Comment (EC1) · Kirsten Elger (Editor) · 24 Feb 2020

Dear Joao et al, please take the opportunity to answer the referee's comments in the interactive discussion of ESSD. This might avoid a second round of reviews which I would suggest based on the current recommendations of the referees. As your deadline to submit the revised version of the manuscript and your authors responses ends today, I have asked Copernicus to extend it until the 9 March. This should allow you to add your answers in the public discussion.

I would also like to address the following:

At the moment, the data publication (http://doi.org/10.5880/ICGEM.2019.006) has the same title as the manuscript. This is unfortunate. because readers might get confused

having two references with similar titles. At GFZ Data Services, we recommend a title describing your data, which is perfect with the title you have chosen. However, would it be possible to slightly adapt the title of the manuscript? I could imagin to simply add "Description of" to the title (Description of multi-approach gravity field models from SWARM GPS data) or similar. Would this be possible?

Many thanks and best regards,

Kirsten Elger

———————————————

---

## Author Comment (AC2) · 20 Apr 2020

Please be advised that the complete reference for Jäggi et al. (2020) is:

Jäggi, A., Meyer, U., Lasser, M., Jenny, B., Lopez, T., Flechtner, F., Dahle, C., Förste, C., Mayer-Gürr, T., Kvas, A., Lemoine, J.-M., Bourgogne, S., Weigelt, M., and Groh, A.: International Combination Service for Time-variable Gravity Fields (COST-G) – Start of operational phase and future perspectives, in: IAG Symposia Series, in press, https://doi.org/10.1007/1345_2020_109, 2020.

The DOI number has just been assigned.

João Encarnação

---

## Author Response (AR1)

First of all, we thank all reviewers for their insightful comments, which undoubtedly allowed us to improve the quality of the manuscript. To aid their review of our rebuttal, all references to Figures, Tables and Sections are click-able links to the manuscript below this document.

We would like to report the following changes which have modified somewhat our analysis, although the interpretations and conclusions remain largely unchanged.

In deriving the time series of epoch-wise spatial RMS (Figures 9, 12, 13, 14, 16 and 18 in the original version) we erroneously reported that the positive trend in Gravity Recovery And Climate Experiment (GRACE)'s spatial variability is not caused by long-term trends (e.g. ice-loss in polar regions). To better check our procedure, we explicitly removed all trends from the coefficients before computing these epoch-wise statistics. We verified that the positive trend in GRACE's spatial variability was not longer present. Evidently, our original check of looking at the standard deviation and grid mean is unsuitable to determine the cause of the positive trend.

As a consequence, the GRACE's spatial RMS is flat (cf. Figure 6, 9 and 13 of the revised version) and is not larger than the spatial RMS of the Swarm difference w.r.t. GRACE climatological model over land (not shown). It is therefore difficult to convey our original messages comparing the geophysical signal amplitude with the Swarm/GRACE residual when the signal is defined by the climatological model. In Sections 3.3 and 3.4, for this reason, we decided to compare Swarm to GRACE directly, which allows for deviations from the climatological model represented in both GRACE and Swarm to improve the statistics (lowering slightly the amplitude of the difference), thus better illustrating the accuracy of the Swarm oceanic/land signal, as well as the effect of smoothing.

We reiterate that this modification changes mostly the amplitudes of the epoch-wise GRACE spatial RMS (previously derived from the GRACE climatological model including its trends) and only slightly lowers the amplitude of the difference between Swarm and GRACE (because large-scale geophysical deviations from the climatological model are observed by both GRACE and Swarm). Our intention is to illustrate how the ability of Swarm to observe the same processes as GRACE evolves with time, which requires the Swarm/GRACE residual to be compared with the GRACE signal. With the detrended climatological model, the previously used Signal-to-Noise Ratio (SNR) ratio defined as epoch-wise spatial RMS would only substantiate a higher smoothing than what we practically observe to be sufficient (as demonstrated in Section 3.5).

The following changes are not shown in the version of the manuscript below (where deletions are in red and additions are in green). This is because they either make the document more difficult to read or are not related to the scientific content of the manuscript:

- Christoph Dahle's affiliation has been corrected;

- CK Shum's affiliation has been updated;

- all plots are somewhat different, but generally do not change any conceptual interpretation (unless otherwise noted in this rebuttal). The changes relate to the use of a different $C_{2,0}$ time series and the addition of the Swarm and GRACE data up to September 2019;

- the figure captions have been revised to be in better agreement;

- some figures are now shown as sub-figures, as listed in our response to Reviewer's 1 first comment;

- Figure 1 and Table 5 from the original version have been removed;

- the URLs in the data access statement have been updated, as well as the reference to the published data;

- the funding statement has been updated to include relevant institutions (in the acknowledgements section).

The following elements changed significantly and we consider that a justification is beneficial to understand the results:

- Figures in Sections 3.3 and 3.4, i.e. Figures 6 to 14 and 16 to 17 (Figures 9 to 19 in the original version), have changed because we detrend the models before computing the spatial variability. We:

- no longer plot the climatological model but the full (detrended) GRACE signal epoch-wise RMS (we maintained the corresponding line as yellow, since these two lines are closely related);

- replace the Swarm difference w.r.t. the GRACE climatological model with the full (detrended) Swarm signal epoch-wise RMS (in blue);

- replace the GRACE difference w.r.t. the GRACE model with the Swarm/GRACE residual epoch-wise RMS (red).

- Figure 26 (Figure 28 in the original version) has a much less obvious equator signature (had we not reduced the maximum value of the colour bar). This is because of the same reason as the previous item and to be in agreement to what we have done in Sections 3.3 and 3.4.

- the values of Table 7 (Table 8 in the original version) have changed as a result of using a different $C_{2,0}$ coefficient; this is particularly the case of regions near the poles and related to the mean basin Equivalent Water Height (EWH).

**R1 comment:** General Review: [...] One general comment is that the figures are not of high quality – the legends and axes are all very difficult to read. Further, there are many places where multiple figures could be combined into a single figure with subpanels. This would likely increase readability.

**Response:** we have re-plotted all figures and re-arranged Figures 4-6 in the original version in a single figure (now Figure 3). We have also increased the font size.

**R1 comment:** Section 2.6.1 – It is unclear to me why the authors decided to use this approach for C20. They state that the C20 coefficient available from CSR is only available at the GRACE epochs. However, the same group at CSR produces a monthly 5x5 gravity field solution from SLR, which includes as estimate of C20. These estimates are produced in even calendar month intervals. Why not use this?

**Response:** The reviewer's suggestion would require two time series of $C_{2,0}$ to be considered: one for GRACE (the TN-11, since this time series is purposely produced to correct GRACE's $C_{2,0}$ and available at the epochs of the GRACE solutions) and another for Swarm (the monthly 5x5 gravity field solution from Satellite Laser Ranging (SLR), since it is available continuously at similar epochs to the Swarm models). This is certainly a possibility but we prefer to avoid any influence on the results resulting from the differences between these two time series. We also agree our initial approach is questionable given the reviewer's and other available options; for this reason we have considered the 7-day time series provided by Goddard Space Flight Center (GSFC), which introduces negligible interpolation errors.

Consequently, We have replaced the paragraphs related to the empirical $C_{2,0}$ model with:

... we selected the $C_{2,0}$ 7-day time series from Loomis et al. (2019), since the necessary interpolation introduces negligible deviations. We are not advocating that the considered $C_{2,0}$ time series is in any way superior to other solutions, e.g. Cheng et al. (2011) (which is only available at the middle of calendar months) or Cheng and Ries (2018) (which is only available for epochs compatible with the GRACE monthly solutions); we have selected it purely under the consideration it was the most technically convenient option for our needs.

**R1 comment:** Line 323: Typo: "if" should be "to"
**Response:** Corrected.

**R1 comment:** It is unclear what the authors mean by "These periods drive the orbital inclination of the GRACE satellites...". Perhaps you mean the other way around, i.e., that the orbital inclination drives the tidal aliasing periods? This would be more appropriate.
**Response:** Corrected.

**R1 comment:** Line 386: Typo – "which is less straightforward"
**Response:** Corrected.

**R1 comment:** The authors state it is a mystery as to why the oceans have larger errors than over land. Is this really true? A comparison between Figure 14 and Figure 16 reveals that the error over the ocean is only about $\approx 25\%$ higher than that over land. It is true that the signal to noise ratio is much lower over the ocean – this is why larger smoothing radii are required – but that seems mostly due to the lower signal amplitudes rather than the higher noise values. Have you tried assessing errors as a function of latitude rather than over different geophysical domains? I wonder if errors actually scale with latitude (Figure 28 would support this), and errors are slightly larger over the ocean simply because of sampling bias as a function of latitude?

**Response:** The reviewer has identified correctly a series of statements that failed to convey our message clearly. Our message is closely related to the comments of the reviewer, except that we intended to discuss the 25% difference between land and ocean disagreement with GRACE, which we cannot properly justify. We investigated the reviewer's suggestion that sampling these statistics over ocean and land areas may inadvertently introduce a bias associated with higher errors in the equatorial region (where ocean areas are most frequent) and plotted the same statistics for the areas between and outside the tropics. As it can be seen in Figure 15 of the revised version, the differences in the Swarm residual are much less pronounced in the tropical/non-tropical case than in land/ocean case. We have revised this paragraph as follows (and moved it to a more appropriate location):

> The results presented in Figures 11 to 14, illustrate that the Swarm Gravity Field Models (GFMs) are unable to resolve the gravity signal in the oceanic regions at spatial lengths comparable to land areas. We observe that the discrepancy with respect to GRACE over the ocean is roughly 25% larger than over land. We do not have a definitive explanation for this, other than the ionospheric activity may corrupt more significantly the estimated gravity field parameters over the oceans since away from land areas there is very little gravity signal to capture. In other words, the natural gravity variations over land are of sufficient amplitude to dominate the errors, at least enough to drive our statistics.

> The higher accuracy over land could be explained by the ionospheric activity affecting mainly ocean areas, since those are mostly located along the equator (e.g. the Pacific ocean). Masking the land areas could therefore remove the large land signals associated with hydrology and leave mostly the errors in the equatorial oceans. To test this hypothesis, we masked the Swarm/GRACE residual along tropical and non-tropical regions, as illustrated in Figure 15. It is clear that Swarm observes the tropical regions, which include regions with strong gravitational variations such as the Amazon basin and vast ocean areas in the Pacific, in as good agreement as the non-tropical regions. We note that the deep ocean regions are not the complementary of the land regions (i.e. the two domains do not cover the whole Earth, cf. Section 2.6.2) and it should not be expected that their spatial RMS is proportionally larger than the tropical/non-tropical regions, which are of comparable amplitude between themselves and complementary.

**R1 comment:** Line 652: typo – "out" should be "our"

**Response:** Corrected.

**R2 comment:** As a methodological paper, the paper somewhat lacks a hypothesis. It is clear that by combining many analysis we will have a smoother result. What is expected from the outset? That could have been described better. Which is a pity since they seem to have added interesting new methods. What message is conveyed in view of other LEO missions that could be used for gravity retrievals? Should one have different orbits, instruments, what did we learn now for the next LEO mission?

**Response:** Our intention is not to discuss methodology, but rather describe the combined Swarm gravity field models. We list numerous references describing the methodology followed by the diverse institutes involved in this activity, but we limit their discussion to high-level overviews and comparisons.

The discussion of other/future LEO missions, orbits or instruments is outside the scope of the current study. We have added the following sentence to Section 1 in order to clarify our intentions:

...this manuscript aggregates a series of studies and analyses that, respectively, motivate our processing choices and demonstrate the capabilities of the combined Swarm models to observe mass transport processes at the surface of the Earth on a monthly basis, in a way that is superior to any of its individual models.

**R2 comment:** The big issue for this reviewer is whether the authors were well-advised to submit to ESSD. ESSD focuses on "original research data (sets), furthering the reuse of high- quality data of benefit to Earth system sciences". Here, the focus is clearly on the methodology of generating the data and neither its use nor reuse, and I guess other journals are more appropriate.

**Response:** We do not regard this manuscript as focusing on methodology and we believe we have chosen the correct journal to describe our data. The sentence immediately after the one the reviewer quote states: "The editors encourage submissions on original data or data collections which are of sufficient quality and have the potential to contribute to these aims." We believe the combined Swarm models we describe in this article can benefit the Earth system sciences and we believe we demonstrate that, as well as to what extent it can be done.

Finally, these models are funded (indirectly) by the European Space Agency (ESA) under the Swarm data exploitation program run by the Swarm Data, Innovation and Science Cluster (DISC) and are part of ESA's operational products. As such, our main objective is to illustrate the capabilities of the Swarm models in order to correctly frame the expectations of future users of these data.

**R2 comment:** While the authors motivate their study with the need of the community to rely on data sets for studying "glacial cycles and long-term trends", the GRACE-GRACE-FO gap is 10 months and this is the period where these data will be relevant, in addition to few monthly gaps. It seems like a huge effort and the groups are to be congratulated, but they don't show what Earth Science applications will be enabled now that were not possible. We don't learn from their results for the understanding of processes. In their words, the "consequences of the 10-month gap" are not outlined and it is not clear what we gain.

**Response:** We agree that we unnecessarily omitted additional motivations for the geophysical community to consider Swarm gravity field models. For this reason, we have added the following paragraph:

The measurement of Earth's gravitational changes with Swarm is further motivated by i) the need to increase the accuracy of global mass estimates in order to properly quantify global sea-level rise and ii) the opportunity to provide independent estimates of temporal variations of low-degree coefficients, in particular related to $C_{2,0}$ and $C_{3,0}$, which are weakly observed by GRACE.

However, being a data description paper, we disagree that this manuscript should contain detailed analysis on possible applications for these data. In spite of this, we provide illustrations of how the Swarm data observe mass changes during the GRACE gap in Section 3.5. Our efforts to interpret these illustrations are limited because we regard that type of activity to be outside the scope of a data description paper.

**R2 comment:** The other very major problem for this reviewer is that apparently no independent validation is provided, except comparisons to GRACE. We don't have GRACE them for the gap, but the authors could compare their low-degree results to satellite laser ranging (SLR) solutions, e.g. for single coefficients or for the ocean mass change.

**Response:** We believe there is more than enough published evidence that GRACE is adequate as independent validation for our purposes. We chose not to conduct the SLR study because such analysis would only bring any new insights during GRACE (/GRACE-FO) gaps (when the Swarm quality is stable) and only limited to the lowest degrees. We also regard our manuscript to be already quite extensive and additional analysis would most likely not be beneficial to our main message. The analysis of the $C_{2,0}$ coefficient is a research topic on itself (that's the reason we simply replace that coefficient in GRACE and Swarm with an SLR estimate), well outside our objectives for the current manuscript.

**EC1 comment:** At the moment, the data publication (http://doi.org/10.5880/ICGEM.2019.006) has the same title as the manuscript. This is unfortunate. because readers might get confused having two references with similar titles. At GFZ Data Services, we recommend a title describing your data, which is perfect with the title you have chosen. However,

170       would it be possible to slightly adapt the title of the manuscript? I could imagine to simply add "Description of" to the title (Description of multi-approach gravity field models from SWARM GPS data) or similar. Would this be possible?

**Response:** We have changed the title as suggested.

[revised manuscript text omitted]

---

## Author Response (AR2)

Dear editor,

We have difficulty identifying which discussions are missing in our rebuttal about general or serious objections both reviewers made. For that reason, we list below those that we interpret as general or serious and re-address them in an effort to mitigate any omission in our original rebuttal.

We took this opportunity to update a reference in line 95 and replace a duplicate term in line 265.

Best regards.

João Encarnação et al.

**Reviewer 1**

*General Review: [...] One general comment is that the figures are not of high quality – the legends and axes are all very difficult to read.*

We replotted all figures and increased the font size.

*Further, there are many places where multiple figures could be combined into a single figure with subpanels.*

We re-arranged Figures 4-6 into a single figure (now Figure 3).

*It is unclear to me why the authors decided to use this approach for C20.*

We followed the reviewers' suggestion and resorted to SLR-derived C20. We choose the model kindly provided by Dr. Bryant Loomis from GSFC because that model is on a weekly basis and meets our needs to interpolate over GRACE/GRACE-FO and Swarm epochs.

*The authors state it is a mystery as to why the oceans have larger errors than over land. Is this really true?*

We have followed the reviewers suggestion to look at the latitudinal dependence of the errors and added figure 15 to demonstrate it does not change significantly.

**Reviewer 2**

*As a methodological paper, the paper somewhat lacks a hypothesis. What is expected from the outset? That could have been described better.*

Our hypothesis is that the Swarm models describe mass transport on a monthly basis at the 1500km spatial scale over land with accuracy comparable to GRACE (at the same spatial scales). It is mentioned in the abstract as well as in the conclusions.

*What message is conveyed in view of other LEO missions that could be used for gravity retrievals?*

The Swarm satellites are not dedicated gravimetric satellites. They are equipped with sensors that have particularities that make that application possible but less than ideal. Even with those characteristics, we are unaware of other non-dedicated LEO satellites that are able to observe Earth's temporal changes in gravity. Additionally, going deep into this discussion does not address any characteristic of the models we are presenting, which is our main objective.

*Should one have different orbits, instruments, what did we learn now for the next LEO mission?*

The topics the reviewer suggests is well outside a data description article. They rely exclusively on simulation (i.e. no real data) and requires numerous assumptions (e.g. sensor noise characteristics and background force models errors).

*The big issue for this reviewer is whether the authors were well-advised to submit to ESSD. [...] Here, the focus is clearly on the methodology of generating the data and neither its use nor reuse, and I guess other journals are more appropriate.*

We disagree this is a methodological paper, quite far from that. We mention methodologies in order to illustrate how the data was produced. The data's use and reuse is implicitly to be determined by the audience of this article, who resort it to learn their particularities and the assumptions leading to their production.

*While the authors motivate their study with the need of the community to rely on data sets for studying "glacial cycles and long-term trends", the GRACE-GRACE-FO gap is 10 months and this is the period where these data will be relevant, in addition to few monthly gaps.*

We recognised that the motivation for the Swarm models was lacking and added the application to Sea-level studies and the improvement of the low-degree coefficients.

*It seems like a huge effort and the groups are to be congratulated, but they don't show what Earth Science applications will be enabled now that were not possible.*

Swarm is not a dedicated gravimetric mission. Any model derived from Swarm's data is invariably going to be of less quality than any dedicated mission. For the time being, GRACE-FO is providing that data and Swarm cannot compete with it. We took great care in presenting our results in the context of the GRACE/GRACE-FO data quality exactly to illustrate this fact to the users of the data. Swarm gravity field models may not enable new Earth Science applications (as far as we know), but it is providing independent data that can be used for validating models derived from dedicated missions. We demonstrate this in a few hydrological basins and illustrate a few deviations from the climatological model that are observed by GRACE-FO and Swarm. We essentially show that GRACE, GRACE-FO and Swarm are observing the same Earth.

*We don't learn from their results for the understanding of processes. In their words, the "consequences of the 10-month gap" are not outlined and it is not clear what we gain.*

These are examples of applications for our models, to be undertaken by the user community, as was done by (e.g.) Meyer et al. (2019b), although different models were used.

*The other very major problem for this reviewer is that apparently no independent validation is provided, except comparisons to GRACE. We don't have GRACE them for the gap, but the authors could compare their low-degree results to satellite laser ranging (SLR) solutions, e.g. for single coefficients or for the ocean mass change.*

We are unaware of monthly global gravity field models, other than those produced from GRACE/GRACE-FO. We choose to restrict our analyses to those in the article to limit the length of the paper. The ocean mass changes has already been object of study in Luck (2018), as mentioned in Section 3.4.2. The topic of observing C20 with gravimetric satellites, in particular, stands on its own; any discussion could not be done in an acceptable way as a subsection on a data description paper. It is a good example of an application for the user community.

*The authors mention only one SLR time series for C20 but that's just one data set and more exist. Others provided C02 timeseries and it is not clear to this reviewer why they rely on GRACE extrapolatons for this.*

We have replaced the C20 with SLR-derived time series, also in response to R1.

*Moreover, the authors, e.g. in Berne or Potsdam, could easily use their Swarm models in SLR range residual analysis as a validation but it is not done.*

This was done by the various colleagues (references in Table 1) pertaining to the quality of their respective kinematic orbits. Those articles are much more adequate for the analyses the reviewer suggests because they are done at the level of observations (kinematic orbits), which are most compatible with the SLR observations. We know the information from those orbits has been correctly synthesised in our models because of the agreement with GRACE.

*And several other validation techniques and data sets were developed for GRACE but nothing is used here. This is somewhat disappointing.*

We could not understand to which validation techniques and data sets the reviewer refers. In any case, as already mentioned, the length of the article discourages additional validations.